# Redox-tunable isoindigos for electrochemically mediated carbon capture

Xing Li [1], Xunhua Zhao[2,3], Lingyu Zhang[1], Anmol Mathur[1], Yu Xu[1], Zhiwei Fang[4], Luo Gu [4], Yuanyue Liu[2] & Yayuan Liu [1]✉

Efficient $CO_2$ separation technologies are essential for mitigating climate change. Compared to traditional thermochemical methods, electrochemically mediated carbon capture using redox-tunable sorbents emerges as a promising alternative due to its versatility and energy efficiency. However, the undesirable linear free-energy relationship between redox potential and $CO_2$ binding affinity in existing chemistry makes it fundamentally challenging to optimise key sorbent properties independently via chemical modifications. Here, we demonstrate a design paradigm for electrochemically mediated carbon capture sorbents, which breaks the undesirable scaling relationship by leveraging intramolecular hydrogen bonding in isoindigo derivatives. The redox potentials of isoindigos can be anodically shifted by >350 mV to impart sorbents with high oxygen stability without compromising $CO_2$ binding, culminating in a system with minimised parasitic reactions. With the synthetic space presented, our effort provides a generalisable strategy to finetune interactions between redox-active organic molecules and $CO_2$, addressing a longstanding challenge in developing effective carbon capture methods driven by non-conventional stimuli.

Carbon capture from stationary emitters or directly from the ambient environment, followed by sequestration or utilisation, is critical to mitigating climate change[1-3]. However, the incumbent wet chemical scrubbing methods for carbon dioxide ($CO_2$) separation are technically and economically challenged by various inherent limitations, including high energy consumption for sorbent regeneration, thermal degradation, complexity in heat integration when retrofitting existing infrastructures, process equipment corrosion, and fugitive emission of volatile toxic sorbents to the environment[4,5]. Alternatively, electrochemically mediated carbon capture (EMCC) has emerged as a promising technology[6-11]. In EMCC, reversible $CO_2$ capture and release is modulated by switching electrochemical potentials. Therefore, they can be operated isothermally at ambient pressure, powered by renewable energy sources, and modularly designed to accommodate the multiscale nature of carbon capture

needs. Among the EMCC mechanisms explored to date, one popular strategy is to use redox-active organic compounds as $CO_2$ carriers (redox-tunable Lewis bases), with quinones being the most studied class of molecules[12-14]. Electro-reduction of these molecules generates nucleophiles that form adducts with electrophilic $CO_2$, which can be later oxidised to liberate pure $CO_2$ while regenerating the sorbents (Fig. 1a). The past two decades have witnessed steady research progress in developing EMCC processes using redox-tunable sorbents[8,10,15-20]. Several bench-scale prototypes have been demonstrated for fixed-bed and flow-based $CO_2$ separations attributed to materials and device-level engineering efforts[17-19]. In contrast, molecular-level design principles for precise sorbent property tuning remain largely unestablished beyond the simplistic method of structural substitution with electron-donating and withdrawing groups, despite their central role in EMCC.

[1]Department of Chemical and Biomolecular Engineering, Johns Hopkins University, Baltimore, MD 21218, USA. [2]Department of Mechanical Engineering & Texas Materials Institute, The University of Texas at Austin, Austin, TX 78712, USA. [3]Macao Institute of Materials Science and Engineering (MIMSE), Faculty of Innovation Engineering, Macau University of Science and Technology, Taipa, Macau 999078, China. [4]Department of Materials Science and Engineering, Johns Hopkins University, Baltimore, MD 21218, USA. ✉e-mail: yayuanliu@jhu.edu

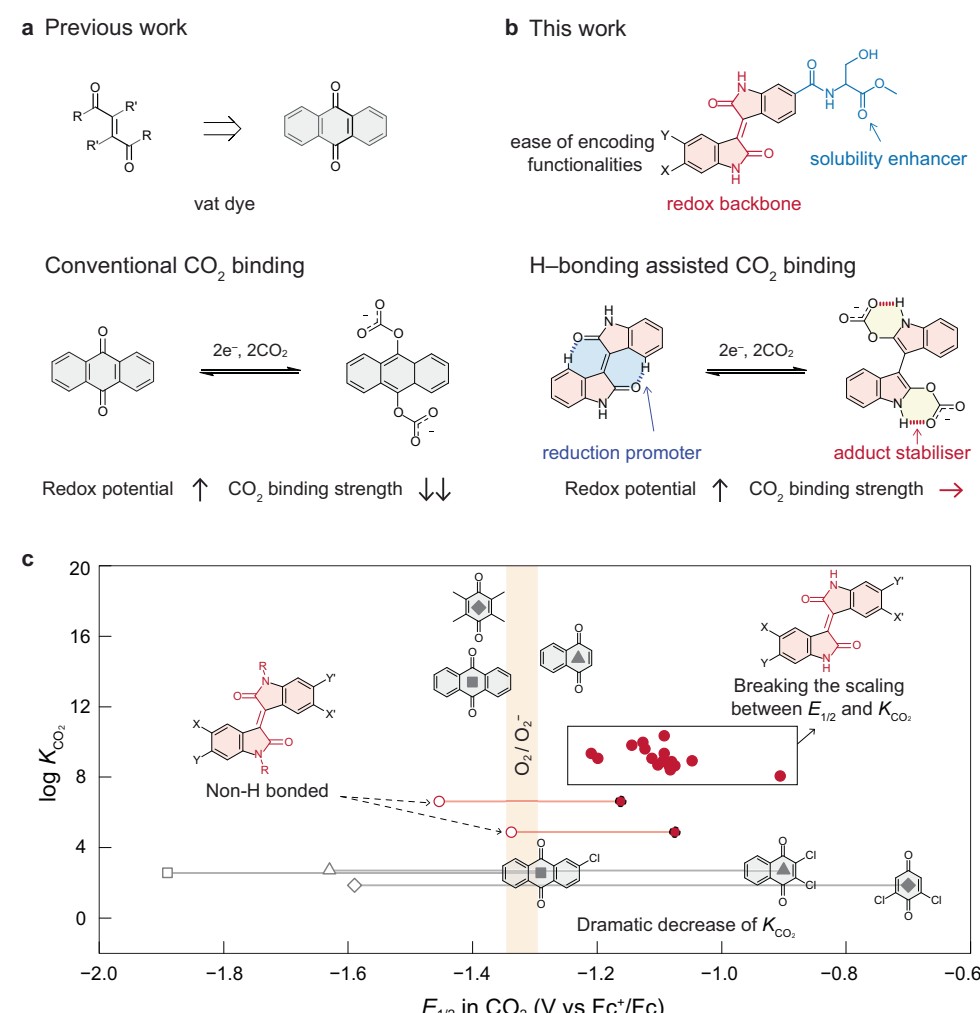

**Fig. 1 | Rational design of bifunctional redox-tunable $CO_2$ sorbents based on isoindigo and their derivatives. a** A universal designing pattern for redox-tunable $CO_2$ carriers containing $\alpha,\beta$-unsaturated 1,4-diketone functionality. In previous designs, $CO_2$ is only bonded to the reduced O centre via carbonate formation, where the binding affinity is sensitive to structural modification. **b** Bifunctional redox-tunable $CO_2$ carrier design based on isoindigo. The secondary functionality of amide allows intramolecular hydrogen bonding, providing an extra handle to stabilise the $CO_2$ adduct, thereby enhancing the $CO_2$ binding strength. **c** A summary of $E_{1/2}$ of typical quinone-based sorbents and isoindigos under $CO_2$, and their $\log K_{CO_2}$ in DMF (filled and empty dots represent the first and second $E_{1/2}$ under $CO_2$, respectively circle, isoindigos; diamond, benzoquinone; triangle, naphthoquinone; square, anthraquinone).

The lack of reliable sorbent chemistry has intrinsically hindered existing EMCC processes. A practical EMCC system usually requires chemical modification of redox-tunable sorbents to improve key properties such as solubility, processability, and stability against impurities. For example, anodically shifting the redox potential will enhance the robustness of activated sorbent against molecular oxygen ($O_2$), a common gas stream impurity, for improved carbon capture efficiency. Nevertheless, the $CO_2$ binding affinity of existing redox-tunable Lewis base sorbents, such as quinones, is highly susceptible to chemical modifications, which decreases dramatically as the redox potential shifts anodically[21–23]. The seminal work from W. L. Bell et al. introduced a method to calculate the $CO_2$ binding constant ($K_{CO_2}$) of activated sorbent, which is widely adopted later to evaluate the $CO_2$ binding strength[12]. As an example, anthraquinone exhibits a two-electron-transfer half-wave potential ($E_{1/2}$) of −1.4 V vs. ferrocenium/ferrocene ($Fc^+$/Fc) in $N,N$-dimethylformamide (DMF) under $CO_2$ and a $\log K_{CO_2}$ of ~13.4 (Fig. 1c). The installation of electron-withdrawing groups (EWGs), such as one chloro group at 2-position, can anodically shift $E_{1/2}$ to −1.25 V vs. $Fc^+$/Fc, yet the $\log K_{CO_2}$ decreased substantially to 2.73 (Fig. 1c and Supplementary Table 1)[21]. A minimum $\log K_{CO_2}$ of ~3.0 in DMF is required to attain a practical efficiency for point source

carbon capture (10% $CO_2$), and the $\log K_{CO_2}$ must be >~5.5 for atmospheric $CO_2$ concentration (400 ppm)[12]. Importantly, it is challenging to overcome the coupling between $E_{1/2}$ and $\log K_{CO_2}$ as it is dictated by the fundamental principle in chemistry that electron deficiency facilitates reduction but in return weakens nucleophilicity[24]. Therefore, to fulfil the practical requirements of EMCC, it is pivotal to develop new classes of redox-tunable sorbents that can break the linear free-energy relationship between redox potential and $CO_2$ binding strength to enable aerobic stability and high $CO_2$ capacity (Supplementary Fig. 1).

Here, through analysing the structures of existing EMCC sorbents, we observe that the most representative quinoid species share a common pattern of $\alpha,\beta$-unsaturated 1,4-diketone (Fig. 1a). Inspired by this structural pattern, we present a class of redox-tunable $CO_2$ carriers based on isoindigo compounds, which can successfully overcome the undesirable coupling between redox potential and $CO_2$ binding strength (Fig. 1b, c). With a molecular library of 21 examples and a combined experimental and computational effort, we show that the $\alpha,\beta$-unsaturated 1,4-diketone in isoindigos plays the role of redox backbone for $CO_2$ binding and the amide groups act as extra docking sites for $CO_2$ complexation via intramolecular hydrogen bonding. This unique bifunctional structural design allows a wide range of chemical

modifications to independently optimise key sorbent properties without sacrificing their abilities for $CO_2$ binding. The isoindigo family culminates in an EMCC system that can operate at mild potentials (around −1 V vs. $Fc^+/Fc$) with $\log K_{CO_2}$ maintained at ~9. The value is five orders of magnitude higher than tetrachloroquinone with alcohol additives[22], which is the state-of-the-art sorbent chemistry with an attempt to break the linear free-energy relationship. Flow-based separation prototypes have also been demonstrated to evaluate the EMCC performance of the isoindigo sorbents, which can achieve $CO_2$ capacity utilisation efficiencies up to ~80% and energy consumptions as low as 127.3 kJ $mol^{-1}$ per $CO_2$ capture/release cycle. With intrinsic $O_2$ stability, high structural tunability, and synthetic feasibility, isoindigos are promising to serve as the next-generation EMCC sorbents. Moreover, this work demonstrates a generalisable strategy to overcome the intrinsic linear free-energy limits in redox-active organic species that can be broadly applied to EMCC and beyond.

## Results

### Redox-tunable $CO_2$ absorption of isoindigo

Isoindigo and its derivatives have been extensively utilised as core building blocks in organic semiconductors[25-28], but have rarely been explored as redox molecules for organic electrodes. We envisage isoindigos bearing $\alpha,\beta$-unsaturated 1,4-diketone functionalities to be redox-active and can complex with $CO_2$ at the oxygen centres in the reduced state. A series of electrochemical experiments was conducted to validate the redox-driven interaction between isoindigo (IId) and $CO_2$. First, we performed bulk electrolysis of IId using 0.25 M lithium perchlorate ($LiClO_4$) in dimethyl sulfoxide (DMSO) as the supporting electrolyte under $N_2$ or $CO_2$ atmosphere (Fig. 2a). Under $N_2$, the dark

dispersion of IId turned greenish at the beginning of the reduction and became a clear yellow solution when the reaction was completed. This suggests the formation of the bisindolidenolate intermediate, which is expected to absorb $CO_2$ as a Lewis base. Interestingly, the yellow solution quickly turned red when purged with $CO_2$, implying the formation of the IId·$CO_2$ adduct. To our delight, $^1H$ and $^{13}C$ nuclear magnetic resonance (NMR) spectra of the crude solution confirm the full conversion of IId into IId·$CO_2$ with negligible side products (Fig. 2b). Characteristic peaks for the carbonate carbon were observed at 176 ~ 177 pm in $^{13}C$ NMR. Besides, the proton on the lactam N shifts upfield to 10.33 ppm in IId·$CO_2$ compared to that of 10.89 ppm in IId, strongly evidencing the formation of intramolecular hydrogen bonding with complexed $CO_2$ (highlighted in red in Fig. 2a). The NMR spectra exhibit two types of amide hydrogens and carbonate carbons, which is probably due to the rotational isomerisation of IId·$CO_2$ (Supplementary Fig. 2). The presence of intramolecular hydrogen bonding in IId·$CO_2$ was further verified by variable-temperature (VT) $^1H$ NMR, 2D $^1H$-$^{13}C$ Heteronuclear Single Quantum Coherence (HSQC) NMR, and Fourier transform infra-red (FT-IR) experiments (Supplementary Figs. 3–5, see Supplementary Note 1 for detailed analysis). The full NMR peak assignment of IId and IId·$CO_2$ is given in Supplementary Fig. 2.

Since the discovery of the EMCC mechanism using quinones in 1988[12,13], to our best knowledge, the widely accepted electrochemically generated quinone-$CO_2$ carbonate adduct is still a proposed structure and has not been confirmed in an EMCC process by non-ambiguous characterisations. This is probably due to the poor stability of the adducts and the transient bonding nature between the reduced sorbent and $CO_2$. On the contrary, crude NMR spectra suggest bulk

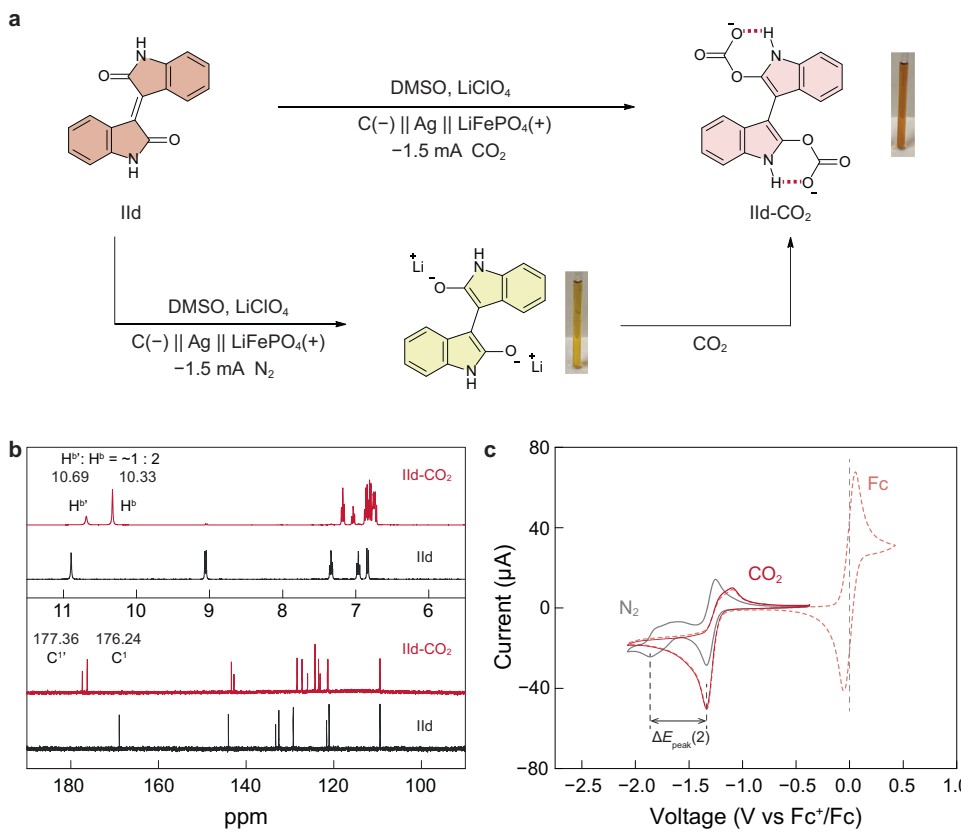

**Fig. 2 | Validating the redox-tunable $CO_2$ absorption of isoindigo. a** Bulk electrolysis of isoindigo (IId) under $N_2$ or $CO_2$ atmosphere. The $CO_2$ adduct can be obtained either by directly reducing IId (40 mM) in DMSO with 0.25 M $LiClO_4$ under $CO_2$ or by reducing IId under $N_2$ followed by purging with $CO_2$. **b** $^1H$ NMR and $^{13}C$ NMR spectra of the crude solution after bulk electrolysis under $CO_2$. The minor

isomer peaks are marked with an asterisk. **c** CV curves of IId (2.5 mM) in DMF using 0.1 M $Nbu_4PF_6$ as the supporting salt under $N_2$ (grey) and $CO_2$ (red) at a scan rate of −50 mV $s^{-1}$ (5 mM ferrocene as the internal reference). $\Delta E_{peak}(2)$ is labelled for the calculation of $K_{CO_2}$.

electro-reduction of IId under $CO_2$ can yield the proposed adduct with full conversion (IId was fully consumed), indicating the high selectivity of this chemistry and the sufficient stability of the adduct, which we postulate to be the result of intramolecular hydrogen bonding. Noticeably, the IId-$CO_2$ solution obtained from bulk electrolysis can be stably stored in ambient air with high water content. After 53 days, <9 mol% of IId-$CO_2$ oxidised back to the neutral IId form (Supplementary Fig. 6).

To confirm the redox activity of IId, we measured its cyclic voltammetry (CV) using 0.1 M tetrabutylammonium hexafluorophosphate (NBu$_4$PF$_6$) in DMF as the supporting electrolyte (Fig. 2c). Under an inert N$_2$ atmosphere, IId exhibits two major redox waves typical to stepwise two-electron transfer. Like quinoid species[13], the two electron transfer steps correspond to the formation of anionic radicals (IId$^-$) and dianions (IId$^{2-}$), respectively. The two reduction peaks under N$_2$ emerge into one in the presence of $CO_2$ with a nearly doubled peak current, indicating chemical interactions between reduced IId and $CO_2$. This behaviour is analogous to the other redox-tunable $CO_2$ sorbents reported previously based on oxygen or nitrogen binding centres, whose reaction with $CO_2$ proceeds via an ECEC mechanism (E, electron transfer; C, chemical reaction)[12,13,20]. As a Lewis acid, $CO_2$ can withdraw the electron density from IId$^-$ to promote the second electron transfer, giving rise to an anodically shifted second reduction wave. Besides, the oxidation peak also shifts anodically and becomes quasi-reversible, corroborating the formation of the IId-$CO_2$ adduct, which requires more energy for $CO_2$ desorption.

This preliminary finding encouraged us to expand the chemical scope of isoindigos and search for more qualified EMCC compounds. Isoindigos can be synthesised through the condensation reaction between 2-oxindoles and isatins. This allows the modular design of redox sorbents, where the two building units can be modified separately and integrated into isoindigos in the last step, thereby addressing the synthetic barriers when engineering EMCC sorbents (Supplementary Note 2). Here, we demonstrate the structural modification of isoindigo at 5, 6, and N-positions with 21 examples (Fig. 3) and examine in detail the interplay between substituent groups and hydrogen bonding on the redox and $CO_2$ binding behaviours of isoindigos to establish the underlying structure-property relationships. The main conclusions are summarised in Fig. 4 and will be discussed in detail in the following sections.

**Tuning the redox potential of isoindigos**

We first examine the redox behaviours of derivatised isoindigos under N$_2$ via CV. Delightfully, all the isoindigos tested display redox couples typical to stepwise two-electron transfer, underscoring the good electrochemical reversibility of these molecules (Fig. 4a, Supplementary Figs. 7–9).

Among the 21 examples of isoindigos, only IId, 6BIId, and 66DBIId exhibit less well-defined shapes for the second redox wave. This is probably due to the rotational isomerisation of the one-electron reduced isoindigo radical anions. As shown in Supplementary Fig. 10, the C = C bond connecting the two oxindole rings becomes a single bond in the radical anion, allowing free rotation of the two rings. This creates rotational isomers and gives rise to shoulder peaks in the second redox process. Adding strong EWGs or substitution groups at the 5-position of isoindigo can create dipole-dipole repulsion or steric hindrance, inhibiting rotational isomerisation and resulting in a more reversible second redox process. Similarly, adding $CO_2$ onto isoindigos also creates dipole and steric hindrance, impeding the rotational isomerisation and leading to more defined CV curves.

Ideal EMCC sorbents shall have redox potentials more positive than the oxygen reduction potential (−1.35 V vs. Fc$^+$/Fc in DMF, Supplementary Fig. 11) to minimise sorbent sensitivity towards O$_2$. With stronger or increasing numbers of EWG introduced to the isoindigo rings, the redox potential exhibits an increasing anodic shift in the

sequence of -F, -Br, -COOMe, -CONHR, and -NO$_2$ substituent groups from mono- to tetra-substitution (Fig. 5a and Supplementary Table 2). Through a close examination of the structure-property relationship, we hypothesise that the anodic shift is attributed to not only the commonly expected electronic state tuning from the EWG substituents but also the intramolecular hydrogen bonding effect (**a** shown in Fig. 4b and c). This is because hydrogen bonding can decrease the electron density of the redox-active oxygen centre to facilitate reduction. The downfield shift of proton at 4-position (H$^a$) in isoindigos with EWG substituents suggests the formation of stronger hydrogen bonding (**a**) (Table 1 and Supplementary Table 2).

To verify the above hypothesis, we show that EWG substituent at 5-position is more effective in facilitating electro-reduction than that at 6-position. This is because the former is in the ortho-position of H$^a$ and more effective in pulling away the electron density, thereby inducing a stronger hydrogen bonding (Fig. 4d). For instance, 55DBIId and 66DBIId exhibit very close $^1$H NMR peaks for H$^b$ (11.11 and 11.10 ppm, Supplementary Fig. 12), suggesting similar degrees of electron deficiency in these two molecules. In contrast, the chemical shift of H$^a$ is 9.32 and 8.99 ppm for 55DBIId and 66DBIId, respectively, clearly indicating a stronger hydrogen bonding (**a**) in the 5-substituted species. Therefore, $E_{1/2}$(IId/IId$^-$) of 55DBIId (−1.09 V vs. Fc$^+$/Fc) is more positive compared to 66DBIId (−1.12 V vs. Fc$^+$/Fc). This trend is consistent for all examples in our isoindigo family (Supplementary Table 2, e.g., 5BIId vs. 6BIId and 6MCIId vs. 5NIId vs. 5N6MCIId) and is further confirmed with DFT calculations (details *vide post*).

Counterintuitively, adding electron-donating groups (EDGs) such as methoxy shows a negligible influence on $E_{1/2}$(IId/IId$^-$), as evidenced by IId vs. 55DMIId (both show $E_{1/2}$ of −1.29 V vs. Fc$^+$/Fc, Table 1), and 5NIId vs. 5M5NIId (both show $E_{1/2}$ of −1.08 V vs. Fc$^+$/Fc, Supplementary Table 2). This is likely due to the charge-transfer effect that lowers the energy level of the molecule, offsetting the electronic effect from EDGs. Specifically, isoindigo species are electron acceptors (n-type organic semiconductors)[28], where charge transfer can be induced between the electron-deficient isoindigo rings and the electron-rich methoxy group. UV-vis absorption spectra suggest an optical bandgap of 1.90 - 1.98 eV for most isoindigos with or without chemical modification (Supplementary Table 2 and Supplementary Figs. 13–15). However, the bandgaps drop to 1.78 and 1.71 eV for methoxy substituted 55DMIId and 5M5NIId, manifesting charge transfer in these two compounds.

**Breaking the scaling relationship between redox potential and $CO_2$ affinity**

After understanding the effect of molecular structures on the redox potentials of isoindigos, we further study their $CO_2$ binding behaviours. All isoindigos in this work exhibit anodically shifted potential for the second electron transfer process under $CO_2$, confirming their ability to form $CO_2$ adducts upon electro-reduction. Importantly, through the combined effect of EWGs substitution and hydrogen bonding (**a**) discussed above, all isoindigos with unsubstituted H$^b$ display $E_{1/2}$ values anodic to oxygen reduction under $CO_2$, implying favourable stability of isoindigo sorbents against O$_2$.

In all previous reports on redox-tunable $CO_2$ sorbents, anodically shifted redox potential always comes with significantly diminished $CO_2$ binding affinity (Supplementary Table 1). However, another key finding of this work is that hydrogen atom (H$^b$) on the lactam-N of isoindigos can induce intermolecular hydrogen bonding with the complexed $CO_2$ molecule to thermodynamically stabilise the $CO_2$ adduct (Fig. 4e). As a result, regardless of $E_{1/2}$, the log $K_{CO_2}$ values of isoindigos with unsubstituted H$^b$ were found to be relatively constant (Table 1 and Supplementary Table 2, see Supplementary Note 3 for details on $K_{CO_2}$ calculation), suggesting the high tolerance of such redox carriers to a wide range of chemical modifications.

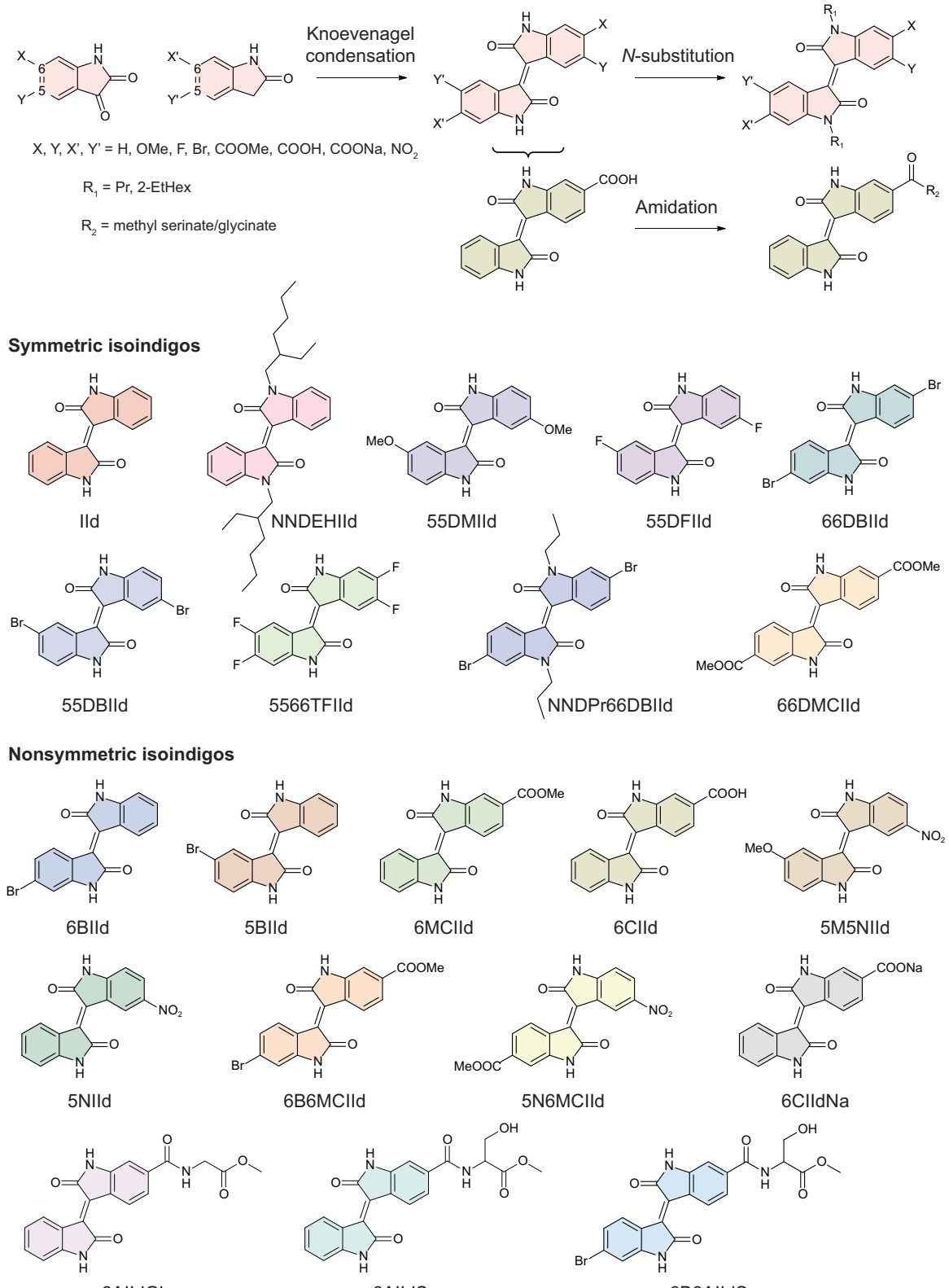

**Fig. 3 | Modular synthesis of redox-tunable isoindigo sorbents for EMCC.** Functional groups can be pre-installed onto the precursors of isoindigos and various isoindigos can be obtained through Knoevenagel condensation of the precursors. Amino acid ester can be further installed through amidation reaction with carboxylic acid modified isoindigos.

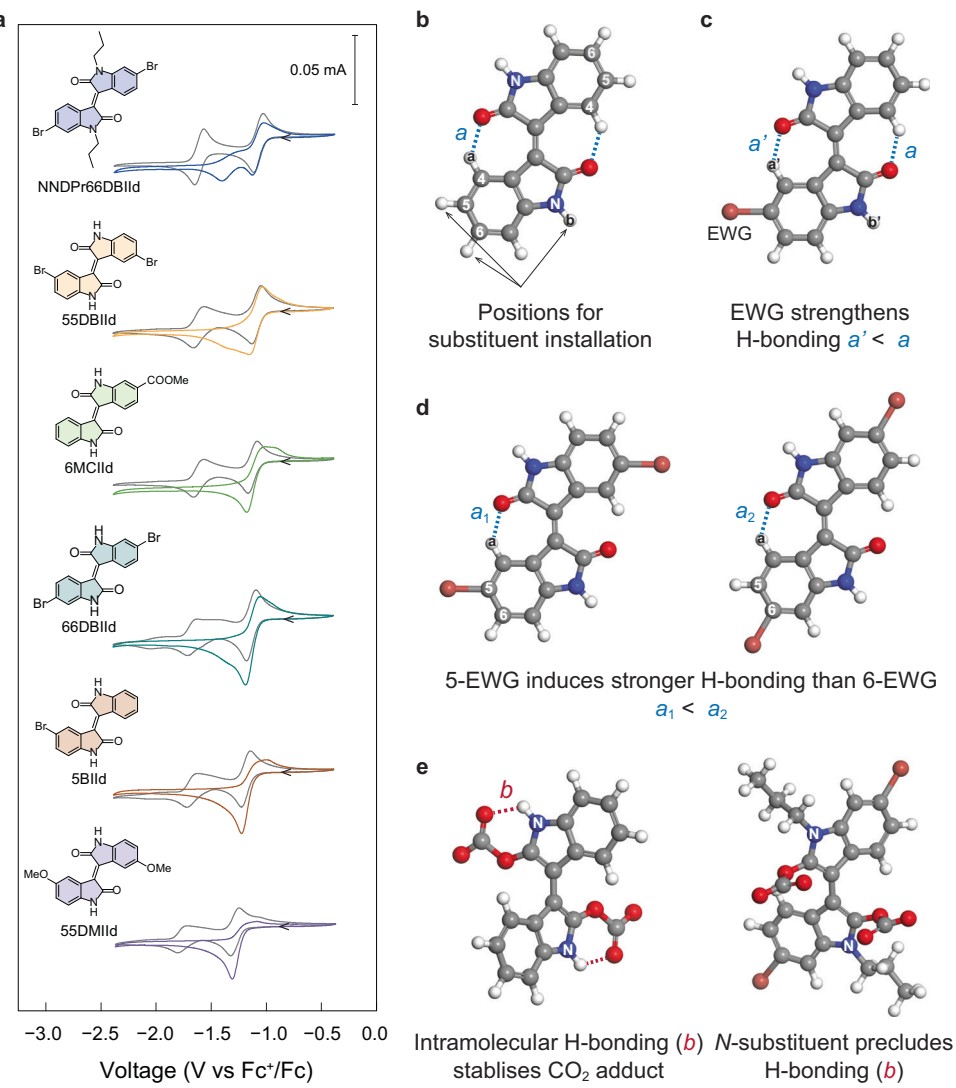

**Fig. 4 | Structure-property relationships of redox-tunable isoindigo-based CO₂ sorbents. a** CV of various isoindigos using 2.5 mM compound in DMF with 0.1 M NBu₄PF₆ under N₂ (grey) or CO₂ (coloured). The CV curves were recorded at a scan rate of −50 mV s⁻¹ at 298 K. **b** DFT-optimised structure of IId; positions for introducing substituent groups are labelled, which are at 5, 6, and lactam *N*-position; intramolecular hydrogen bonding (*a* indicates bond length) in the neutral state is shown as blue dashed lines. **c** DFT-optimised structure of 5BIId suggests EWG can strengthen the hydrogen bonding (*a*). **d** DFT-optimised structures of 55BIId (left) and 66DBIId (right) showing that EWG substituent at 5-position is more effective in strengthening the hydrogen bonding (*a*). **e** DFT simulation showing that intramolecular hydrogen bonding (*b*) indicates bond length, red dashed lines) occurs in the CO₂ adduct, stabilising the CO₂ binding. Colour of atoms: grey: C; red: O; blue: N; brown: Br; white: H.

To further demonstrate the role of hydrogen bonding (*b*) on CO₂ adduct stabilisation, we synthesised two examples with *N*-substitutions to eliminate this hydrogen bonding (NNDPr66DBIId and NNDE-HIId). Compared with 6,6'-dibromo substituted 66DBIId (log $K_{CO_2}$ = 9.59), the CO₂ binding constant of *N*-alkylated NNDPr66DBIId (log $K_{CO_2}$ = 4.48) shows a dramatic decrease of five orders of magnitude (Table 1). A similar phenomenon was observed between NNDEHIId (log $K_{CO_2}$ = 6.62, Supplementary Table 2) and IId (log $K_{CO_2}$ = 9.34). Besides, we designed an isoindigo bearing a carboxylic acid group (6CIId), which serves as a free proton donor to disrupt the intramolecular hydrogen bonding (*b*). As a result, the CO₂ binding ability of reduced 6CIId almost diminished with a low log $K_{CO_2}$ of 2.87 (Supplementary Table 2).

To visualise our success in breaking the scaling relationship between redox potential and CO₂ binding affinity, we plot the change in $K_{CO_2}$ relative to unmodified IId against $E_{1/2}$ of the first electron transfer under N₂ (Fig. 5b). Upon installing EWGs, $E_{1/2}$ can be effectively shifted from −1.29 V to −0.97 V vs. Fc⁺/Fc. Nevertheless, the $K_{CO_2}$ values

exhibit minimal changes within only one order of magnitude, as long as the lactam-N is unsubstituted to facilitate intramolecular hydrogen bonding. Furthermore, linear fitting shows a negligible correlation ($R^2 < 0.1$), underlining that our molecular design strategy can indeed break the scaling relationship between redox potential and CO₂ binding affinity.

We further tested the bimolecular rate constant ($k_{bimolecular}$) for the reaction between isoindigo radical anion and CO₂ (Supplementary Figs. 16–19, Supplementary Table 3, see Supplementary Note 3 for details in measurement). With intramolecular hydrogen bonding (*b*) in the CO₂ adduct, 6MCIId, 6AIIdSer, and 66DBIId display a similar rate constant of 22.5–18.6 M⁻¹ s⁻¹. However, the *N,N*-disubstituted NNDPr66DBIId exhibits a decreased rate constant of 3.3 M⁻¹ s⁻¹, manifesting that intramolecular hydrogen bonding (*b*) is also conducive to CO₂ complexation kinetics.

At a CO₂ concentration of 20% or 10%, the CV curves of isoindigos usually exhibit a positively shifted second reduction peak compared to that under N₂; however, most do not completely

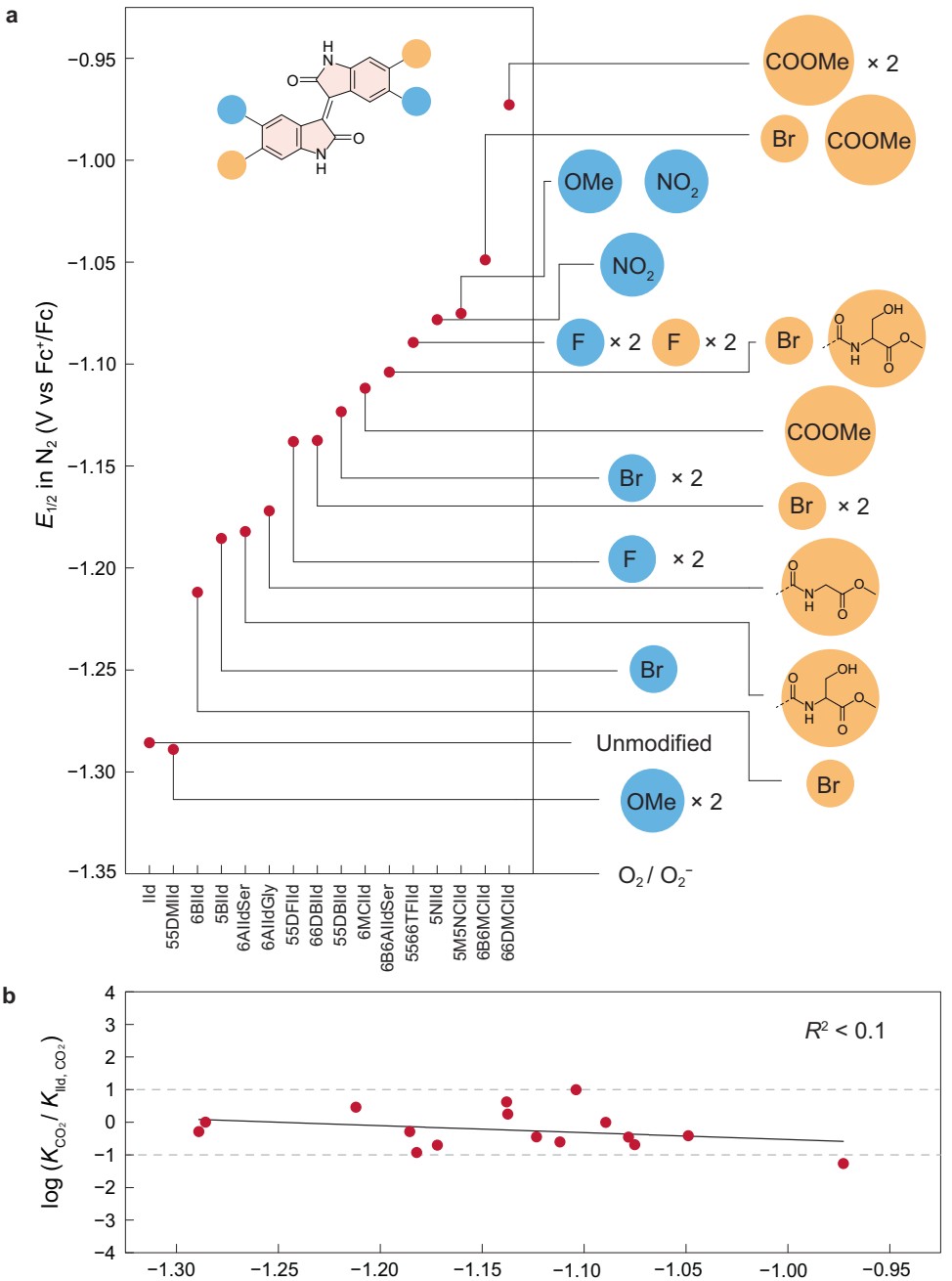

**Fig. 5 | The substituent effect on the redox potentials and $CO_2$ binding abilities of isoindigos. a** The effect of substituent group on the first electron-transfer half-wave potentials under $N_2$. **b** The plot between the first electron transfer half-wave potential under $N_2$ and the change in $CO_2$ binding constant relative to unmodified isoindigo. There is no linear correlation ($R^2 < 0.1$) between the relative $CO_2$ binding constants and the half-wave potentials.

emerge into the first (Supplementary Fig. 7 and 8). The phenomenon is attributed to the kinetic competition between the chemical transformation of $IId^{\bullet-}$ into $[IId\text{-}CO_2]^{\bullet-}$ ($r = k_{bimolecular}[CO_2][IId^{\bullet-}]^{29}$, where $[CO_2]$ and $[IId^{\bullet-}]$ are the concentrations of $CO_2$ and $IId^{\bullet-}$) and the electrochemical reduction of $[IId\text{-}CO_2]^{\bullet-}$. Higher $CO_2$ concentrations lead to higher $r$, which facilitates the formation of $[IId\text{-}CO_2]^{\bullet-}$ and ultimately leads to a single, merged reduction peak. Correspondingly, we tested the CV of 6MCIId at different scan rates and indeed observed the gradual merging of the two reduction peaks with slower scans (Supplementary Fig. 20). Therefore, the separation between the two cathodic peaks under low $CO_2$ concentrations can serve as a qualitative indicator for $CO_2$ complexation kinetics.

As a final note, $CO_2$ complexation can be frustrated when introducing highly strong EWGs or strong hydrogen bonding acceptors to isoindigo (5N6MCIId and 6CIIdNa). Detailed analysis is included in Supplementary Note 4 and Supplementary Fig. 21.

## Detailed investigation on the role of hydrogen bonding

Distinct from previously reported redox-tunable $CO_2$ carriers, isoindigos possess intramolecular hydrogen bondings in both inactivated (neutral) and activated (reduced) forms, accounting for their unique EMCC properties. Therefore, we selected the seven most representative structures from our library and carefully compared various parameters such as NMR spectra, density functional theory (DFT)-optimised bond lengths, redox potentials, and $CO_2$ binding constants

**Table 1 | The interplay between substituent groups and intramolecular hydrogen bondings in isoindigos and the corresponding impacts on redox potentials (V vs. Fc⁺/Fc) and CO₂ binding**

| Isoindigos | ¹HNMR (Hᵃ) (ppm)[A] | ¹HNMR (Hᵇ) (ppm)[A] | Bond length of a (Å)[B] | Bond length of b (Å)[B] | Bond length of c (Å)[B] | $E_{1/2}$(IId/IId•⁻) in $N_2$ (V vs Fc⁺/Fc) | log $K_{CO_2}$ |
|---|---|---|---|---|---|---|---|
| IId | 9.06 | 10.89 | 1.963 | 1.954 | 1.461 | −1.29 | 9.34 |
| 55DMIId | 8.85 | 10.69 | 1.953 | 1.97 | 1.463 | −1.29 | 9.05 |
| 5BIId | 9.07 | 10.96 | 1.962 | 1.957 | 1.463 | −1.19 | 9.05 |
| 5BIId (EWG) | 9.31 | 11.05 | 1.936 | 1.943 | 1.467 | −1.19 | 9.05 |
| 55DBIId | 9.32 | 11.11 | 1.939 | 1.946 | 1.469 | −1.09 | 9.33 |
| 6MCIId | 9.07 | 10.94 | 1.960 | 1.954 | 1.954 | −1.12 | 8.89 |
| 6MCIId (EWG) | 9.15 | 11.08 | 1.956 | 1.944 | 1.476 | −1.12 | 8.89 |
| 66DBIId | 8.99 | 11.1 | 1.961 | 1.942 | 1.468 | −1.14 | 9.59 |
| NNDPr66DBIId | 9.05 | NA | 1.948 | NA | 1.486 | −1.07 | 4.87 |

[A] ¹H NMR were recorded on the neutral molecules in DMSO-d₆ using solvent residual peak as the internal reference for calibration. [B] Bond length was obtained from DFT-optimised structures. [C] Results from the non-symmetric oxindole ring with the EWG-substituent.

to investigate the interplay between substituent groups and hydrogen bonding on the thermodynamic properties of isoindigos. Key data are summarised in Table 1 (also see Supplementary Figs. 22 and 23 for DFT-optimised structures). DFT calculation shows that the theoretical first electron transfer potential shifts anodically in the order of IId, 5BIId, 66DBIId, and 55DBIId, consistent with experimental observation (Supplementary Table 4). Besides, DFT-calculated $CO_2$ binding constants ($\log K_{CO_2}$) of the isoindigos agree well with the trend of our experimental results (Supplementary Table 5). DFT-optimised structures further confirm the formation of intramolecular hydrogen bonding (*a*) and (*b*).

The electron density of $H^a$ at 4-position and $H^b$ at N-position can be regarded as indicators of the bond strength of *a* and *b*, respectively. Specifically, [1]H NMR spectra reveal a chemical shift of 9.06 ppm for $H^a$ and 10.89 ppm for $H^b$ in unmodified IId, corresponding to a DFT-optimised bond length of 1.963 Å for *a* and 1.954 Å for *b*. Introducing electron-withdrawing bromo groups at 5-position (55DBIId) downfield shifts $H^a$ to 9.32 ppm and $H^b$ to 11.11 ppm, resulting in an enhanced hydrogen bonding of 1.939 Å for *a* and 1.946 Å for *b*, respectively. Interestingly, the electronics and hydrogen bonding of each oxindole ring can be independently tuned in nonsymmetric isoindigos. Using the nonsymmetric 5BIId as an example, the chemical shifts of $H^a$ and $H^b$ are 9.07 and 10.96 ppm at the non-substituted side, corresponding to a DFT-optimised bond length of 1.962 Å for *a* and 1.957 Å for *b*, which is very close to IId. In contrast, $H^a$ and $H^b$ shift downfield to 9.31 and 11.05 ppm at the bromo-substituted side, corresponding to an enhanced hydrogen bonding of 1.936 Å for *a* and 1.943 Å for *b*.

Moreover, introducing the same EWG at different positions modulates the strength of hydrogen bonding (*a*) differently. For example, with the same dibromo-substitution, the chemical shift of $H^a$ in 66DBIId (8.99 ppm) is upfield to that of 55DBIId (9.32 ppm). This results in a weakened hydrogen bonding (*a*) of 1.961 Å in 66DBIId than that of 1.939 Å in 55DBIId, explaining the more negative reduction potential of 66DBIId (−1.14 V vs. $Fc^+$/Fc) than 55DBIId (−1.09 V vs. $Fc^+$/Fc). The above results strongly suggest that, in addition to the electronic effect of substituent groups, intramolecular hydrogen bonding (*a*) is also vital in facilitating the reduction of isoindigo.

Although decreasing the electron density of isoindigos by introducing EWGs at 5,6-positions can effectively facilitate their reduction, counterintuitively, the reduced isoindigos exhibit negligible decay in $CO_2$ affinities, underscoring the importance of intramolecular hydrogen bonding (*b*). Using DFT calculation, we found that the nucleophilicity of the oxygen centre indeed weakens when EWG is introduced, as suggested by the increased length of the carbonate C−O bond (*c*). For instance, the bond length of *c* on the oxindole ring with −COOMe group is increased by 1.2 pm compared to that on the non-substituted ring in 6MCIId. However, as mentioned above, hydrogen bonding (*b*) strengthens with stronger or increasing number of EWG substituents to keep the bond length of *c* nearly constant. Moreover, the DFT-optimised structure suggests that hydrogen bonding (*b*) is precluded in NNDPr66DBIId, and the carbonate bends out-of-plane to the reduced isoindigo rings due to steric repulsion. Thus, the bond length of *c* in NNDPr66DBIId is increased by 1.8 pm compared to 66DBIId, giving rise to a significant drop in $K_{CO_2}$ by almost five orders of magnitude.

The collective information above confirms our key conclusions. First, hydrogen bonding (*a*) can reduce the electron density at the redox centre and facilitate reduction. Second, hydrogen bonding (*a*) can be tuned by substituent groups, where EWG at 5-position is more effective than 6-position to shorten the bond length of *a* and hence facilitate reduction. Third, hydrogen bonding (*b*) stabilises the complexed $CO_2$ when EWGs are introduced.

It is important to note that the effect of intramolecular hydrogen bonding on $CO_2$ complexation has been briefly studied in prior works using quinones. However, it was observed that hydrogen bonding occupies the $CO_2$ binding sites of quinones and diminishes the ability for $CO_2$ capture[14,30]. Therefore, our work presents the first demonstration that intramolecular hydrogen bonding can facilitate $CO_2$ adduct formation and break the intrinsic linear free-energy relationship of EMCC chemistries. This is attributed to the unique chemical structure of isoindigo that allows free rotation of the oxindole rings in the reduced state as supported by DFT simulation, breaking the intramolecular hydrogen bonding (*a*) to create space for $CO_2$ complexation, which is further enhanced by the intramolecular hydrogen bonding (*b*) through the amide functionality.

## Finetuning the properties of isoindigos

Breaking the correlation between chemical modification and $CO_2$ binding affinity greatly enhances the degree of freedom in finetuning sorbent properties. For instance, by installing EWGs such as methyl carboxylate, $E_{1/2}$ of 66DMCIId under $CO_2$ can be positively shifted by 300 mV compared to that of unmodified IId to impart $O_2$ stability, while the $\log K_{CO_2}$ only drops slightly (from 9.34 to 8.07). Besides, 66DMCIId is almost insoluble in organic solvents such as DMF (solubility < 2.5 mM), suggesting its potential as absorbent electrodes in fixed-bed EMCC devices.

Unmodified IId has a moderate solubility in DMF ( ~ 230 mM), which needs to be improved for practical use in flow-based EMCC systems[7]. To facilitate chemical functionalisation, we introduced a carboxyl group to the 6-position of isoindigos, which can be easily connected with amino acids through amidation reactions. As a proof of concept, we utilised glycine methyl ester and serine methyl ester as solubility enhancers and three nonsymmetric isoindigos were prepared (6AIIdGly, 6AIIdSer, and 6B6AIIdSer). 6B6AIIdSer features halogen substitution on one oxindole ring to tune redox potentials and amino acid ester functionalisation on the other oxindole ring to enhance solubility. To our delight, the solubilities of 6AIIdGly, 6AIIdSer, and 6B6AIIdSer increase to 606, 830, and 568 mM in DMF, respectively (Supplementary Table 6), likely due to the enhanced molecular interaction between the polar functional groups (amide and carboxylate) and DMF solvent. This is supported by the fact that 6AIIdSer is more soluble than 6AIIdGly due to the additional hydroxyl group from the serine moiety.

Conventional amine scrubbing sorbents have raised environmental concerns due to their biotoxicity[31,32]. Here, we show that introducing amino ester functionalities into isoindigos substantially improves their biocompatibility with mammalian cells. Unmodified IId shows a $LC_{50}$ (lethal concentration that causes 50% cell death) of 11.2, 50.4, and 15.8 µg ml[−1] for NIH3T3/GFP mouse fibroblasts, U2OS.EGFP human osteosarcoma cells, and MCF10A human breast epithelial cells, respectively, after 48 h cell culture (Supplementary Fig. 24 and 25). In comparison, the serinate-modified counterpart does not display clear toxicity under concentrations up to 100 µg ml[−1] for NIH3T3/GFP and U2OS.EGFP, and has a significantly improved $LC_{50}$ of 89.4 µg ml[−1] for MCF10A.

## Effects of electrolytes, oxygen, and water

Before evaluating the $CO_2$ capture performance of isoindigos in EMCC devices, we assessed the influence of electrolytes and common gas stream impurities such as water and $O_2$ on their $CO_2$ binding properties. Using 55DBIId as an example, the electrochemical behaviours remained almost unaffected up to a high $O_2$ content (16% $CO_2$ and 20% $O_2$, Supplementary Fig. 26). Under $N_2$, the CV curves of 55DBIId remain reversible even at a high water content of 10 vol% (Supplementary Fig. 27). The declining peak current is caused by the decreasing isoindigo solubility with increasing water content. However, the expected $CO_2$ release at approximately −1 V vs. $Fc^+$/Fc gets suppressed in the presence of 10 vol% water, possibly due to the involvement of pH-swing process under high water content that requires higher energy input for $CO_2$ release.

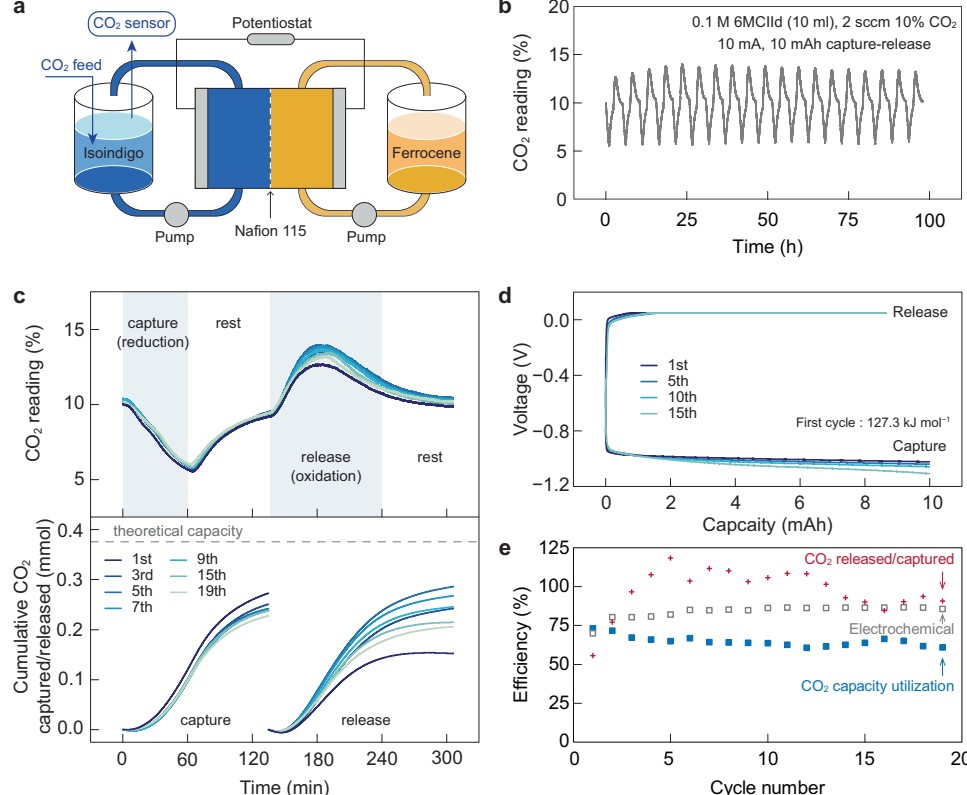

**Fig. 6 | Evaluating the performance of 6MCIId in the flow-based EMCC prototype. a** Schematic of the flow-based EMCC prototype. **b** $CO_2$ reading at the exit of the sorbent tank over 19 repeating capture/release cycles for ~100 h of operation. **c**, The $CO_2$ reading of selected capture/release cycles overlaid, with the cumulative amount of $CO_2$ captured/released in each cycle relative to the theoretical capacity. Lighter colours represent later cycles. The shaded regions indicate the capture, rest, release, and rest steps. For $CO_2$ capture, 6MCIId was reduced at 10 mA for 60 min followed by a 75 min rest. For $CO_2$ release, the adducts were oxidised at 10 mA to 0.05 mV followed by a ~120 min constant voltage hold, and finally rested for another 75 min. **d** Selected voltage-capacity curves for the 1st, 5th, 10th, and 15th capture/release cycle. **e** The $CO_2$ capacity utilisation efficiency (blue squares), release/capture efficiency (red crosses), and electrochemical efficiency (empty grey squares) of the system. The liquid sorbent was composed of 10 ml 0.1 M 6MCIId in DMF with 0.25 M $NaClO_4$ as the supporting salt. The sorbent tank was filled with plastic beads and purged with 10% $CO_2$ at a flow rate of 2 standard cubic centimetres per minute (sccm). On the opposite side, a Fc tank was used to balance the charge, which was filled with 20 ml 0.1 M Fc in DMF with 0.25 M $NaClO_4$ as the supporting salt, 4 mM ferrocenium tetrafluoroboronate ($FcBF_4$) to facilitate Fc oxidation, and 10 mM 6MCIId to mitigate sorbent crossover.

We also studied the influence of supporting salt on the redox and $CO_2$ binding behaviours of isoindigos (Supplementary Fig. 28). The most prominent effect comes from the choice of cation, where reducing the size of cation leads to anodically shifted reduction potential, as is explained by the electrostatic interaction between cation and reduced isoindigo. Smaller alkaline cations exhibit stronger Lewis acidity, allowing tighter binding with reduced isoindigo to facilitate electro-reduction. Therefore, a more acidic supporting salt cation can further enhance the robustness of isoindigo against $O_2$. Nevertheless, it may also slow down the $CO_2$ complexation kinetics due to the competition between cation and $CO_2$ for binding with reduced isoindigo.

### Evaluating the isoindigo sorbents in flow-based EMCC prototypes

Based on the CV peak potentials for $CO_2$ capture and release, we estimated the theoretical minimum energy requirement for $CO_2$ separation using isoindigo sorbents, which ranges from 9.8 to 27.6 kJ mol$^{-1}$ $CO_2$ (Supplementary Fig. 29). It is noteworthy that these molecules show very close onset potentials for $CO_2$ capture and release such that the theoretical energetics calculated from CV peak potentials can be overestimated compared to previous calculations using onset values[17]. As a proof of concept, we evaluated their intrinsic capability for reversible $CO_2$ capture and release in a flow-based EMCC prototype reported by us previously (Fig. 6a)[17]. Detailed testing conditions are provided in the Supplementary Information.

Figure 6b shows the cyclic capture-release performance of 6MCIId using 10% $CO_2$ (balanced by $N_2$) as the gas feed. The $CO_2$ reading curves of each cycle are overlaid in Fig. 6c, where the cumulative $CO_2$ captured/released is obtained by integrating these curves. For each cycle, 6MCIId was reduced at 10 mA for 60 min, and the decrease in $CO_2$ concentration at the gas outlet confirmed carbon capture. The current was then stopped for 75 min, allowing the $CO_2$ reading to gradually return to the baseline as the reduced isoindigo fully reacted with $CO_2$. The $CO_2$ adduct was subsequently oxidised following a constant current-constant voltage (CC-CV) protocol, and the continuous increase in $CO_2$ concentration above 10% indicated $CO_2$ desorption. Afterward, the current was set to zero again to ensure the complete release of the oversaturated $CO_2$ from the sorbent electrolyte. The oxidation-reduction profiles of the flow system are shown in Fig. 6d. By integrating the voltage-capacity curves, the electrical energy consumption under 10% $CO_2$ is estimated as 127.3 kJ mol$^{-1}$ $CO_2$ in the first cycle and 142.5 ± 8.2 kJ mol$^{-1}$ $CO_2$ over the first 16 cycles (Supplementary Fig. 30), which is comparable to other carbon capture technologies[18–20,33–38].

Figure 6e summarises the three key metrics commonly used for evaluating EMCC performance. $CO_2$ capacity utilisation, defined as the amount of $CO_2$ captured relative to the theoretical value (one $CO_2$ per electron), shows an average of 65% over 19 cycles, which is competitive against the state-of-the-art quinone-based sorbent reported recently[19]. The release/capture efficiency, defined as the ratio of the total amount

of $CO_2$ released and captured in each cycle, is averaged to be 97%, indicating the good reversibility of the sorbent. Considering a relatively constant electrochemical (Coulombic) efficiency of the EMCC prototype at ~84%, the major loss should be attributed to two factors: (1) our CC-CV protocol where the $CO_2$ adduct was not fully oxidised; (2) the crossover of sorbents and counter electrolytes caused by membrane swelling, which limits all current nonaqueous redox-flow electrochemical systems. 6MCIId was also evaluated at a higher percentage of $CO_2$ removal (Supplementary Fig. 31).

In addition to 6MCIId, we evaluated the performance of other isoindigo sorbents such as 55DBIId (Supplementary Fig. 32), 66DBIId (Supplementary Fig. 33), and 6B6AIIdSer (Supplementary Fig. 34). 55DBIId achieved an average $CO_2$ utilisation efficiency of up to ~80% and an average release/capture efficiency of ~80%. Using 66DBIId, we studied $^1$H NMR of the crude sorbent electrolyte after 11 capture/release cycles over 50+ hours (Supplementary Fig. 35). The spectrum suggests the high stability of 66DBIId after cycling and also the severe crossover issue of the ferrocene counter electrolyte (three times the concentration of 66DBIId in the sorbent tank), which explains the decay in electrochemical capacity of current EMCC prototypes. Besides, we can recover 66DBIId with 87% yield from the sorbent electrolyte after cycling, corroborating the robustness of the sorbent. In addition, we found ~24% of 66DBIId isomerised into cis-66DBIId, supporting our hypothesis on the rotational isomerisation of reduced 66DBIId discussed earlier (Supplementary Fig. 10). Nevertheless, the reduction of both 66DBIId and its cis-isomer yield the same activated $CO_2$ sorbent, which we believe does not affect the long-term stability of the EMCC prototype.

To our delight, 6B6AIIdSer exhibits a much lower oxidation potential for $CO_2$ release, likely due to its higher solubility. Moreover, 6B6AIIdSer shows excellent cycling stability with negligible voltage decay over >40 cycles and ~200 h of operation with a capacity degradation rate of 2% (Supplementary Note 5).

We further tested the EMCC performance of 6MCIId using simulated flue gas (10% $CO_2$ + 3% $O_2$ balanced in $N_2$) (Supplementary Fig. 36). The cell can run stably over ~90 h with reduction voltage maintained above −1.3 V, which minimised the parasitic oxygen reduction reaction. A $CO_2$ capacity utilisation efficiency of ~50% was achieved with a near unity $CO_2$ release/capture efficiency.

Finally, we evaluated the $CO_2$ capture capability of 6MCIId under low $CO_2$ concentration and its $CO_2$ release capability under pure $CO_2$. Using 1% $CO_2$ with 0.3% $O_2$ as the feed, we observe an early-stage energy consumption of 224.2 kJ mol$^{-1}$ $CO_2$ and a single pass $CO_2$ removal of >90% (Supplementary Fig. 37). This suggests that our intramolecular hydrogen bonding strategy is effective in improving $CO_2$ affinity for low-concentration $CO_2$ capture. In another experiment, the $CO_2$ capture-release behaviour under 100% $CO_2$ headspace was quantified using mass flow metre (Supplementary Fig. 38). Under conditions similar to low-concentration $CO_2$ capture, we show an early-stage energy consumption of 143.7 kJ mol$^{-1}$ $CO_2$ captured and 13.6 kJ mol$^{-1}$ $CO_2$ released, respectively.

In this study, we focus on exploring the fundamental chemistry of isoindigos as redox-active $CO_2$ carriers and their potential to overcome the linear free-energy relationship that limits the structural modification of EMCC sorbents. The flow-based prototype in this work is, however, not an ultimate design for practical systems but a proof-of-concept demonstration to evaluate the performance of isoindigo at the lab scale. We believe future efforts can substantially improve the performance by optimising the electrolytes, electrodes, and membranes of EMCC devices.

## Comparison of methods for estimating $CO_2$ binding constants

As a final note, in this manuscript, we estimated the $K_{CO_2}$ of isoindigos using the prevalent method adopted for quinones, bipyridines, and benzyl thiolate[12,21,22], providing an equitable comparison with previously reported redox-active $CO_2$ carriers. Accordingly, we assume that the two-electron-reduced isoindigo binds to one molecule of $CO_2$ and calculate the $K_{CO_2}$ based on $\Delta E_{peak}(2)$ under pure $N_2$ and $CO_2$ atmosphere, respectively (Supplementary Note 3). This method, therefore, eliminates the kinetic effects in CV under lower $CO_2$ concentrations. Alternatively, we recorded the CV of IId and 6MCIId under various $CO_2$ concentrations (Supplementary Fig. 39) and fitted the relationship between $\Delta E_{peak}(2)$ and $CO_2$ concentration (Supplementary Fig. 40 and Supplementary Note 7). Similar $K_{CO_2}$ values were obtained for IId and 6MCIId on the order of $10^{12}$, further confirming the high tolerance of EWG in isoindigo structural motifs for strong $CO_2$ binding. However, the fitting quality was unsatisfactory, with low $R^2$ values and unreasonable number of binding sites. Therefore, we took the former method to calculate $K_{CO_2}$ for all isoindigos reported in this work.

## Discussion

In summary, we demonstrate the rational design of a class of bifunctional redox-tunable $CO_2$ carriers based on isoindigo and their derivatives. The unique intramolecular hydrogen bonding in isoindigo moieties enables a wide range of chemical modifications to facilitate electro-reduction, tune solubility, and preclude parasitic reactions without compromising their $CO_2$ binding ability. With coupled experimental and computational studies, we provide an in-depth analysis of the structure-function relationships of isoindigos as EMCC sorbents. Compared to existing EMCC sorbents, isoindigo compounds have the following advantages: 1) a nearly constant log $K_{CO_2}$ of ~9 with high tolerance to chemical modifications; 2) facile synthesis with the ease of encoding functionalities; 3) highly tunable redox potentials and solubilities; 4) improvable biocompatibility. In addition to flow-based EMCC, we envisage that isoindigos can also find applications as solid adsorbents in fixed-bed systems, due to their descent charge mobility and the abundant methods in synthesising and processing isoindigo-based polymers developed by the community of organic semiconductors. The work paves the way for engineering more reliable EMCC systems by breaking the fundamental barriers of the scaling relationship between redox potential and $CO_2$ binding strength when designing redox-tunable $CO_2$ sorbents.

## Methods
### Synthesis of 6MCIId
To a mixture of isatin (1.47 g, 10 mmol) and methyl 2-oxindole-6-carboxylate (1.91 g, 10 mmol) in acetic acid (50 ml) was added 37% HCl solution (0.5 ml). The mixture was heated at reflux for 1 day under Argon atmosphere. The mixture was cooled to room temperature, filtered, and washed with water, ethanol, and ethyl acetate. The solid was dried in the vacuum oven at 60 °C for 15 h to afford a dark red powder (2.93 g, 92%). $^1$H NMR (400 MHz, DMSO) δ 11.08 (s, 1H), 10.94 (s, 1H), 9.15 (d, J = 8.4 Hz, 1H), 9.07 (d, J = 8.0 Hz, 1H), 7.57 (dd, J = 8.4, 1.7 Hz, 1H), 7.38 (td, J = 7.7, 1.2 Hz, 1H), 7.34 (d, J = 1.6 Hz, 1H), 7.02 – 6.94 (m, 1H), 6.85 (d, J = 7.3 Hz, 1H), 3.87 (s, 3H). $^{13}$C NMR (101 MHz, DMSO) δ 168.68, 168.62, 165.57, 144.70, 143.91, 135.73, 133.47, 131.95, 131.79, 129.83, 129.06, 125.68, 121.98, 121.51, 121.28, 109.69, 109.29, 52.29.

### Electrochemical measurements
Electrochemical measurements were performed with a BioLogic VSP potentiostat from BioLogic Science Instruments. Cyclic voltammetry (CV) utilised a glassy carbon electrode (3 mm diameter) as the working electrode, a platinum wire as the counter electrode, and a silver wire as the quasi-reference electrode, with ferrocene as the internal reference. In a standard CV test, isoindigo (2.5 mM) was dissolved in anhydrous DMF with 100 mM NBu$_4$PF$_6$ as the supporting electrolyte. CV were typically recorded at a scanning rate of −50 mV s$^{-1}$ with a cut-off potential from −1.5 to 0.5 V (prior to ferrocene calibration). To examine the effects of ionic species on the redox behaviour of the sorbent

molecules, supporting electrolyte salts including 100 mM LiClO₄, NaClO₄, KClO₄, NBu₄ClO₄, sodium triflate (NaOTf), or sodium bis(trifluoromethanesulfonyl)imide (NaTFSI) was employed, respectively. The electrochemical data were gathered and analysed by EC-lab V11.50.

## Flow-based EMCC prototype

In a standard setup, a scintillation vial (20 ml) with a septum cap, serving as the sorbent tank, was continuously purged with $CO_2$ feed gas (balanced with $N_2$) at a controlled flow rate using an Alicat mass flow controller. Simulated flue gas conditions were mimicked using a 10% $CO_2$ and 3% $O_2$ mix, balanced by $N_2$. $CO_2$ levels were continuously monitored at the gas outlet using an infra-red-based $CO_2$ sensor (SprintIR-W 100%), with data recorded via Labview 2021. To minimise mixing time in the overhead space, the tank was filled with plastic beads from McMaster-Carr. The Fc tank was maintained without air exposure. Sorbent and Fc electrolytes were circulated at a flow rate of 10 ml min⁻¹ through a commercial flow cell (Scribner) by a two-channel peristaltic pump (Masterflex). The flow cell incorporated two graphite plates with 5 cm² interdigitated flow fields, pressing against two pieces of carbon paper electrodes (Sigracet 28 AA) on each side of the graphite electrodes. A Nafion 115 membrane, flanked by polypropylene sheets, was placed between the carbon electrodes. The cell was sealed by Kalrez fluoropolymer elastomer gaskets (0.02-inch thick). $CO_2$ capture was conducted in constant current mode, while release followed a constant current/constant voltage protocol: the adduct was first oxidised at a constant current until reaching a cut-off voltage and then held at that voltage until the current value fell below 5% of the original constant current). For experiments using 5 cm² flow fields, the $CO_2$ capture-release was cycled at a current of 10 mA with a cut-off potential of 0.05 V in the release process. For experiments using 25 cm² flow fields, the $CO_2$ capture-release was conducted at a current of 50 mA with a cut-off potential of 0.3 V in the release process. The cycling protocols of each experiment are provided in the corresponding figure captions. For experiments using 6MCIId sorbent, we added 10 mM 6MCIId and 4 mM ferrocenium tetrafluoroboronate (FcBF₄) into the CE tank to mitigate sorbent crossover and facilitate the reduction of the oxidised ferrocenium in the counter electrolyte, respectively. For less soluble 55DBIId and 66DBIId, the sorbent was dispersed in dimethylacetamide (DMAc) to form a slurry catholyte.

## Reporting summary

Further information on research design is available in the Nature Portfolio Reporting Summary linked to this article.

# Data availability

The data generated or analysed during this study are included in the manuscript and its Supplementary Information. The main data generated in this study are provided in the Supplementary Information/ Source Data file. Data are also available from the corresponding author upon request. Source data are provided with this paper.

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

## Acknowledgements

We acknowledge support from the Johns Hopkins University, the Scialog program sponsored jointly by Research Corporation for Science Advancement and the Alfred P. Sloan Foundation, with additional support from Climate Pathfinders Foundation (grant #28438 X.L.), and the David and Lucile Packard Foundation. X.Z. acknowledges FDCT of Macau SAR (grant#0024/2022/ITP).

## Author contributions

X.L. and Ya.L. conceived of the project and designed the experiments. X.L. conducted the experiments and analysed the data. X.Z. performed the DFT simulations under the supervision of Yu.L. L.Z. helped with the CV experiments and solubility test. A.M. helped analyse the carbon capture results. Y.X. performed the reaction rate tests and analysed the data. Z.F. conducted the biocompatibility tests under the supervision of L.G. Ya.L. supervised the whole project. X.L. and Ya.L. co-wrote the manuscript. All authors discussed the results and revised or commented on the manuscript.

## Competing interests

The authors declare no competing interests.
