## [Peer Review File · Nature Communications]

Redox-Tunable Isoindigos for Electrochemically Mediated Carbon CaptureREVIEWER COMMENTS

Reviewer #1 (Remarks to the Author):

Review of: Redox-Tunable Isoindigos for Electrochemically Mediated Carbon Capture

Noteworthy Results of the Paper:

Quinones are the most studied class of molecule for redox-active sorbent based electrochemical CO₂ capture systems however they are limited by their reactivity towards O₂. It has been shown that quinones are limited by a linear free energy relationship which shows that quinones with redox potentials positive of O₂/O₂⁻ do not have a high enough CO₂ binding constant for effective capture from either flue gases or for direct air capture.

In this exciting paper the authors tackle this issue by expanding on previous work by Barlow and Yang (JACS 144, 31, 2022, 14161-14169) (which showed that hydrogen bonding additives stabilise quinone dianions). Here the authors design and synthesise a new type of redox-active sorbent (isoindigos) which are proposed to have strong CO₂ binding due to stabilisation of the CO₂-bound product by H-bonding. This kind of approach goes beyond the work by Barlow and Yang, and avoids the need for volatile additives in a CO₂ capture system. A lot of work has been completed with 21 molecules synthesised and tested.

The central claim of the paper is that the new class of molecules breaks the previously observed scaling relationship, where molecules with more positive potentials bind CO₂ more weakly. If true, this claim would be very interesting. However, I have concerns about the validity of this claim (see below), due to issues with the analysis.

The authors also use some of their isoindigos in a redox-flow setup where they used potentials positive of O₂ reduction and directly measured CO₂ capture and release, demonstrating at least one of their molecules work in ECCS with the presence of O₂ at the bench-top scale, which is exciting.

Are the claims of the study supported?

I see some major issues in the analysis that make me concerned that the claims may not be supported by the data.

1. A key part of the paper is the measurement of CO₂ binding constants for the various isoindigos. I believe the method used by the authors is incorrect, because the peak shift method cannot be used in cases where the two reduction peaks have merged (sometimes known as the "strong binding" regime). In such cases the maximum observable peak shift in the cyclic voltammogram is determined solely by the separation of the two redox peaks under nitrogen. Use of the peak shift equation in these cases will therefore always give very similar binding constants for several molecules (since the calculated binding constants now depends solely on the peak separations under nitrogen, which are generally similar for

the series of molecules). Indeed we see in Table 1 that many of the molecules are predicted to have very similar identical binding constants (all around 9), which is a key claim of the paper.

This criticism is supported by inspection of the authors' data at 20% CO₂ loading in Figure S3. Comparing 55DBIId and 5BIId in Figure S3 (20% CO₂ data), the smaller peak shift of the two molecules is for 55DBIId, suggesting weaker binding. However, the authors calculate and report that 55DBIId has a slightly higher binding constant than 5BIId. As another example, 55DBIId and IId are reported to have very similar binding constants in Table 1, yet their CVs have very different peak shifts under 20% CO₂. All this underscores the point that the method used to calculate the binding constants has been applied incorrectly.

The above points are problematic for the central claims of the paper, which is that the scaling relationship between redox potential and CO₂ binding constant has been broken by the new class of molecules. A key finding of the work is therefore not supported by the data.

I believe a better method is therefore required to obtain the binding constants. E.g. CO₂ concentrations could be varied to lower values to measure reliable peak shifts, and a linear fit of the peak shift equation could be performed at various concentrations. I think if the authors can update their analyses and claims, the quality of work could be improved significantly.

2. I believe there are issues with the interpretation of the NMR spectra in Figure 1b. Two resonances are observed in the range 10-11 ppm for ¹H NMR, and two peaks are observed in ¹³C NMR, close to 180 ppm. The authors write: "The NMR spectra exhibit two types of amide hydrogens and carbonate carbons, which is due to the competition between the hydrogen bonding and Li⁺ complexation (Fig. 2c)." In my opinion this is unlikely to be the correct assignment/explanation, since these species would undergo fast chemical exchange in solution, and would most likely not be resolvable. This raises the question of what is actually forming here. It appears there are two different products present. A full assignment of the entire NMR spectrum for ¹H and ¹³C should be given. DFT calculation of chemical shifts from the already developed structural models could be helpful here.

3. Related to the above point, the authors mention that: "Besides, the proton on the lactam N shifts upfield to 10.33 ppm in IId-CO₂ compared to that of 10.89 ppm in the neutral IId, strongly evidencing the formation of intermolecular hydrogen bonding with complexed CO₂ (highlighted in red in Fig. 2a)." I disagree with this statement. Hydrogen bond formation of an N-H group would be expected to cause a shift in a positive direction (due to increased nuclear deshielding), not the reported negative shift change. This again raises the question of what is actually happening here.

This is an important issue to be resolved, since it undermines the central hypothesis that H-bonding is stabilising the bound CO₂ species. Chemical shift calculations with DFT would again be of help here.

4. Ideally the ¹H and ¹³C NMR would be shown in Figure 2b before the reaction with CO₂, as well as after, so a comparison can be made. I could not find ¹³C NMR before the CO₂ reaction in the SI, and I think this should be included as a control experiment (sorry if I missed it somewhere!). The full NMR spectra (the full spectral width) are also not included anywhere as far as I could see, and this should be included so the reader can assess the claims better.

5. The authors write: "On the contrary, NMR spectra suggest bulk electro-reduction of IId under CO₂ can yield the proposed adduct with full conversion" Given the concerns I have about the NMR above, I don't think this statement is well supported. What is the evidence for "full conversion"
6. for the bulk electrolysis tests (Figure 2a,b), can the cumulative charge passed be compared to the theoretical value? This would probe whether or not the reduction of the isoindigo was complete, or partially complete.
7. Page 17. The use onset values for calculating theoretical energy efficiencies seem invalid. At the onset potentials a negligible amount of CO₂ capture would take place. So these calculations appear misleading to me.
8. can the authors comment on why the second redox process under nitrogen often appears to be less electrochemically reversible (e.g. Figure 1d, and Figure 4a), with a larger peak to peak separation and a more irregular shape?
9. a few of the CVs show extra redox events which aren't discussed in the paper such as for IId. In the paper they state "Under an inert N₂ atmosphere, IId exhibits two major redox waves typical to stepwise two-electron transfer within the electrochemical stability window of DMF. Like quinoid species¹³, the two electron transfer steps correspond to the formation of anionic radicals (IId^{•-}) and dianions (IId²⁻), respectively" which I find to be somewhat dubious looking at the CV, there should be some discussion of possible extra events.

10. The results from the redox-flow experiments do clearly show however that 6MCIId captures CO₂ upon reduction and releases it upon oxidation (at times however over 100 % of the CO₂ captured is released which at least should have a small amount of discussion). It would have been interesting to see how the performance compared to one of the isoindigos with a CV under N₂ showing less ideal behaviour.
11. The NMR study they performed did show that when the isoindigo-dianion is exposed to CO₂, carbonate peaks appear in the ¹³C NMR, which is evidence that the isoindigos are capturing CO₂ as quinones do, by forming carbonates. In the paper they also state that "Since the discovery of the EMCC

mechanism using quinones in 1988^{12, 13}, to our best knowledge, the widely accepted quinone-CO₂ carbonate adduct is still a proposed structure and has not been confirmed by non-ambiguous characterisations. This is probably due to the poor stability of the adducts and the transient bonding nature between the reduced sorbent and CO₂" this is not correct, Yang and Barlow (JACS 144, 31, 2022, 14161-14169) have used both NMR and IR previously to show that the dianion of tetrachlorobenzoquinone forms a carbonate when exposed to CO₂

12. Measurements of the reaction rate of the radical with CO₂ assume an ECE reaction scheme, while the CVs in most cases suggest an EEC mechanism. Is this analysis valid, given this issue?

Minor comments on the presentation of the work:

- the inclusion of the biocompatibility work was a bit unexpected. Why is this important? Can some rationale for doing these tests be given for the reader in the main text?
- Figure 1 is a bit hard to interpret for me, even as someone who is familiar with this literature. Is it possible for the authors to simplify it somehow. It seems a strange place to include an equation here.
- Table 1 caption - please clarify if NMR data are for the molecule before (or after) CO₂ reaction.
- in the intro, the following sentence could be rewritten to be grammatically correct: "In EMCC, reversible CO₂ capture and release is modulated by applying electrochemical potentials, thereby promising systems that can be operated isothermally at ambient pressure, powered solely by renewable energy sources, and modularly designed to accommodate the multiscale nature of carbon capture needs."
- Figure 2a. I suggest labelling IId on the figure next to the structure, for clarity.
- Fig. 5a. The connection between the data points with a line is a misleading. The data content in this figure could perhaps be better presented as a table.
- there is a tendency in this paper to have long paragraphs comparing single values against each other which makes it at times quite difficult to follow, take the following sentence, "As an example, anthraquinone exhibits a two-electron-transfer half-wave potential (E_{1/2}) of -1.4 V vs. ferrocenium/ferrocene (Fc⁺/Fc) in N,N-dimethylformamide (DMF) under CO₂ and a log CO₂ of ~13.4. The installation of EWGs, such as one chloro group at 2-position, can anodically shift E_{1/2} to -1.25 V vs. Fc⁺/Fc, yet the logCO₂ decreased substantially to 2.73 (Supplementary Table 1)²¹." This information is actually much more clearly shown in Fig.1, though they don't reference it. Again with comparing values in text "To verify the hypothesis, we show that EWG substituent at 5-position is more effective in facilitating electro-reduction than that at 6-position. This is because the former is in the ortho-position of Ha and more effective in pulling away the electron density, thereby inducing a stronger hydrogen bonding (Fig. 4d). For instance, 55DBIId and 66DBIId exhibit very close ¹H NMR peaks for Hb (11.11 and 11.10 ppm, Table 1), suggesting the degrees of electron deficiency are very similar in these two molecules. In stark contrast, ¹H NMR spectra suggest a chemical shift of 9.32 ppm for Ha in 55DBIId and 8.99 ppm for that in 66DBIId, clearly indicating a stronger hydrogen bonding (a) in the 5- substituted species. Therefore, E_{1/2}(IId/IId^{•-}) of 55DBIId (-1.09 V vs. Fc⁺/Fc) is more positive compared to 66DBIId (-1.12 V vs. Fc⁺/Fc), and this trend is consistent for all the examples in our isoindigo family

(Supplementary Table 2, e.g., 5BIId vs. 6BIId and 6MCIId vs. 5NIId vs. 5N6MCIId). This hypothesis is further confirmed with DFT calculations (details vide post).” I find this paragraph very difficult to follow with having to remember values and then go look at the referenced structures. If this could somehow be turned into a figure the argument I believe would be much clearer.

Reviewer #2 (Remarks to the Author):

1. What are the noteworthy results?

This work presents a new class of molecules, isoindigos, for electrochemically mediated carbon capture (EMCC). Isoindigos have been reported in literature for carbon capture prior to this work as an additive for thermal adsorption-based carbon capture.[1, 2] Isoindigos have also been used as organic semiconductors [manuscript reference 25-28]. This manuscript reports use of isoindigos as organic electrodes for EMCC for the first time. This manuscript does well in presenting isoindigos as a class of molecules for EMCC.

Isoindigos build off observations from the more heavily studied EMCC molecules, quinones, and benefits from the same 1,4-diketone functionality. The biggest benefit of isoindigos over quinones is intramolecular hydrogen bonding from the amide group to the CO₂ adduct. This hydrogen bonding allows for consistent binding potential of isoindigos across the several modified structures evaluated in the manuscript. The hydrogen bonding allows for more flexibility in modifying the isoindigo structure without decrease in CO₂ binding properties, which has been a challenge in other EMCC systems. Modification is desirable to anodically shift the CO₂ binding potential to improve oxidative stability and to improve solubility of the EMCC compound. The several modifications demonstrated in the paper show that this class of molecules are flexible to augmentation which may leave room for further improvement in oxidative stability, solubility, or other desirable features.

2. Will the work be of significance to the field and related fields? How does it compare to the established literature? If the work is not original, please provide relevant references.

The significance of this EMCC work is the expansion beyond the more heavily studied EMCC molecules, quinones. Building off quinone knowledge gives credibility to the work. Showing that intramolecular hydrogen bonding in the isoindigo system stabilizes the CO₂ adduct is important for maintaining high CO₂ binding constant ($\log K_{(CO_2)}$) across a range of half potentials ($E_{(1/2)}$). The hydrogen bonding of the isoindigo system is the most interesting and useful benefit of the isoindigos and will likely lead to more research into this new class of EMCC molecules.

Other systems have attempted to break the linear $\log K_{(CO_2)}$ vs $E_{(1/2)}$ relationship in a variety of ways but appear to have more difficulties than the isoindigo system. Some modifications have been conducted in the quinone system to incorporate hydrogen bonding, breaking the linear $\log K_{(CO_2)}$ vs $E_{(1/2)}$ relationship [manuscript reference 10], but there is less flexibility in designing quinones with hydrogen bonding due to that hydrogen bonding is not being an inherent property of all quinones. Utilization of ionic liquids has aided in hydrogen bonding for quinones, but this may not be ideal for the scaled up EMCC system [manuscript reference 17]. Salts in quinone systems have also been used to shift the potential anodically without reducing binding properties, but like ionic liquids, may not be ideal in the scaled up EMCC system [reference 14]. By using the isoindigo platform, modifications are not

required to achieve the stabilizing hydrogen bonding. This leads to greater flexibility in the full system design for future work.

3. Does the work support the conclusions and claims, or is additional evidence needed?

This work does well in supporting the claim that isoindigos break the linear $\log K_{(\text{CO}_2)}$ vs $E_{(1/2)}$ relationship for EMCC capture. This is an important deviation from other EMCC molecule classes which have attempted to use modifications to anodically shift the $E_{(1/2)}$ but have reduced the $\log K_{(\text{CO}_2)}$ such that it may no longer be suitable for carbon capture.

The manuscript uses intentionally poor modifications, replacing the amide hydrogen with an alkane, to highlight that the amide provides stability to the CO₂ adduct. Characterization with H-NMR confirms the hydrogen bonding solidifying the findings in this manuscript. This characterization technique may be of use to other groups who are seeking to add hydrogen bonding stability to their structures. DFT studies support hydrogen bonding playing a role in stabilization of the CO₂ adduct.

4. Are there any flaws in the data analysis, interpretation, and conclusions? Do these prohibit publication or require revision?

For the isoindigo molecules presented, the minimum CO₂ capture energy reported as 9.8 to 27.6 kJ mol⁻¹, which is fantastic if it can be realized. The actual cell performance was shown to be 131.5 (isoindigo: 6MCIID, 10mA 60min cycles) to 230.7 kJ mol⁻¹ (isoindigo: 6MCIID, 12mA 90min cycles) in 10% CO₂. The deviation between minimum CO₂ capture energy and actual cell performance appears to be common in quinone papers as well; a low minimum CO₂ capture energy is theorized, but experiments show a higher requirement. Total energy cost appears to be related to the current applied as well as capacity utilization as seen from the above two 6MCIID tests at different currents and capacity utilizations. The authors are encouraged to identify other parameters causing the deviation and suggest possible improvements to achieve capture energy closer to theory. A few possible parameters are overpotentials due to cell resistance, solubility of CO₂ in the solvent, CO₂ concentration dependence on capture energy, and CO₂ transport.

The experimental energy reported in the flow cell is the first or second cycle. Presumably this is the best cycle and hides variance and degradation. The manuscript does not discuss an average energy cost or standard deviation which may be beneficial. It would be nice to see a degradation rate which may be possible for the 43 cycles of the 6B6AIIIDSer, but that may not be necessary at this stage.

Comparing the 6MCIID test in 10% CO₂ with the 6MCIID in simulated flue gas (10% CO₂, 3% O₂) it is surprising to see a significantly lower capacity utilization (~20%) in the simulated flue gas case. This is despite CV curves under CO₂ and CO₂ + O₂ for 55DBIId being similar (SI Fig 19). Commentary on the deviation may be appropriate.

In the supplementary information, performance of 55DBIId (SI Fig. 24) and 66DBIId (SI Fig. 25) voltage spikes down below the O₂/O₂^{•-} potential in later cycles. This is a concern suggesting degradation of the isoindigo. While a voltage limit can be set as a practical preventative measure to avoid reaching the O₂/O₂^{•-} potential in operation, understanding the degradation method is desirable and could be addressed in the revised manuscript or in future work.

A possible degradation path could be side reactions with the II^{•-} intermediate radical. The radical could pose similar, albeit slower, degradation pathways as the concerning as the O₂/O₂^{•-} pair. At sufficiently

high CO₂ concentrations the II●- may rapidly react with CO₂ with minimal side reactions. At lower CO₂ concentrations there may be more opportunity for side reactions.

In general, there is a larger concern for the EMCC field. EMCC technologies have been studied since the seminal paper from W. L. Bell in 1988 [manuscript reference 12]. It is an interesting and promising technology for CO₂ capture and thus has been studied by several groups over the years. Despite this, most references do not actually separate CO₂ from a low concentration stream into a higher stream. Many of the references, including this research, capture and release CO₂ into the same stream. Some references had separate tests in a lower concentrated stream and again a test with higher concentration stream, usually 100% CO₂. This allows for determining a thermodynamic difference between the two states but does not take into consideration transport effects between the two states. One reference conducted a separation effectively from a 100% CO₂ stream into a lower concentration stream [manuscript reference 35]. One reference simulated behavior between 1% and 5% CO₂ [manuscript reference 18]. A lack of full cell operation, fully concentrating from low to high CO₂, makes it difficult to evaluate the full limitations of the technology. There is reference provided that is an exception, Scovazzo, 2003 [reference 16]. Scovazzo, separates CO₂ from 500ppm to 100% using a quinone, DtBBQ. They showed that it was possible experimentally but did not provide an energy cost associated requiring CO₂ to be bubbled through the reduced quinone overnight.

When the end goal of this technology is to separate CO₂ from a low concentration to a high concentration, it should be shown to achieve this. The evidence shown in this manuscript and its references suggest that it is possible to separate CO₂ from a low concentration stream (i.e., 10% flue gas) and concentrate it into higher purity stream but the energies associated are not fully understood. For instance, it is likely that there are inefficiencies due to different amounts of CO₂ dissolved in the solvent in the capture stream versus the release stream as is described in the seminal paper from Bell [manuscript reference 12]. Other inefficiencies are not addressed such as resistance or pH when water is dissolved in the solvent or brought in with the feed stream. This will add to the total energy cost of the EMCC device and assist in determining the viability of the carbon capture method. Furthermore, metrics such as CO₂ flux will need to be considered to assess the cost of materials required for scaleup.

If these EMCC technologies are being considered for flue gas operation, understanding how the cell performs at sub 1% CO₂ concentrations are important for achieving high removal (>90%) from flue gas streams.[3] Experimental setups in this manuscript and others referenced have not explored sub 1% concentrations with the exception of Scovazzo, 2003 [reference 16]. At lower concentrations, presumably there will be higher tendency for isoindigos or quinones to exhibit both upper and lower reduction peaks which is less prevalent at higher CO₂ concentrations. This may lower the effective capacity of these EMCCs. Furthermore, the second reduction potentials may be below the O₂/O₂●- potential raising concerns for oxidative stability. Thankfully, in this research, there is a compound presented with O₂/O₂●- stability by possessing a second redox potential equal to the O₂/O₂●- potential of -1.35 vs Fc⁺/Fc in DMF (6CIIId SI Fig. 5). All others have a second redox potential below the O₂/O₂●- potential in N₂.

This manuscript does mention that for direct air capture (DAC) a binding constant logK_(co₂) of 5.5 is required to separate from 400 ppm CO₂ (reference 12). This research does not further address DAC, and other referenced EMCC research does not test cycling behavior and energy costs in DAC conditions. Without further evidence, the viability of this technology for DAC is questionable despite its meeting the required logK_(co₂) binding constant.

Given that isoindigos are new in the EMCC field, lack of a full energy cost analysis on the system level

does not preclude this manuscript from being published. It is hoped that such as full energy cost analysis is forthcoming at some point. In general, this manuscript will help future work to study the different separation use cases expected (Flue gas, DAC, etc.).

There are a couple further questions in the manuscript. First is why biocompatibility is brought up at all in this manuscript? The supplementary information shows that mammalian cell tests were conducted and showed adding -Ser modification allowed for the isoindigo to be more biocompatible than the base isoindigo. Only one sentence was given in the manuscript on biocompatibility along with two words in the conclusion, "good biocompatibility". Why is this important? Is this supposed to show it is safer? Are there concepts that EMCC would be used in a biological system? It is interesting as a curious test, but it is not clear how the result fit in.

No citations are given about the need for biocompatibility nor methodology. Does the 5% CO₂ incubation condition influence the results. Considering that this compound is intended to capture and release CO₂, would a lower or higher CO₂ concentration matter? Would an isoindigo cycling between CO₂ adduct form and CO₂ free form affect the biocompatibility? The authors are suggested to fully explain the purpose for these tests or remove them from the paper. In its current form, the mammalian cell test distracts from the main purpose of the paper which is EMCC.

5. Is the methodology sound? Does the work meet the expected standards in your field?

Yes, the methodology is built upon related EMCC technologies. The methodologies presented provide parameters that can be used to compare across the field. DFT studies support hydrogen bonding playing a role in stabilization of the CO₂ adduct. H-NMR being used as a tool to verify hydrogen bonding further supports the claims presented. The numerous substituent groups studied in the paper provide experimental examples of how different moieties may affect the isoindigo CO₂ binding properties. This provides a good starting point into future isoindigo EMCC research.

6. Is there enough detail provided in the methods for the work to be reproduced?

Yes, I appreciated the details on the reaction setup and yields for each of the isoindigos presented in the paper. If possible, it would be good to understand if side reactions are present, such as the formation of the indigo isomer. For the flow based EMCC prototypes sufficient operating parameters and design setup are provided to replicate.

Some minor point: EWG appears not defined when first used. It might be a good idea to show the O₂ redox potential in one of the figures to pictorially display which compound has a more positive redox potential.

1. Zou, L., et al., Porous Organic Polymers for Post-Combustion Carbon Capture. *Adv Mater*, 2017. 29(37).
2. Zhao, Y., et al., Isoindigo-based microporous organic polymers for carbon dioxide capture. *RSC Adv.*, 2015. 5(121): p. 100322-100329.
3. Brandl, P., Beyond 90% capture: Possible, but at what cost? *International Journal of Greenhouse Gas Control*, 2021. 105: p. 103239.

Reviewer #3 (Remarks to the Author):

Reviewer #4 (Remarks to the Author):

The present study designed and prepared a new class of bifunctional redox-tunable CO₂ carriers based on isoindigo and their derivatives. The unique intramolecular hydrogen bonding in isoindigo moieties can be tuned by chemical modification on different sites, which can promote electron reduction, change solubility and avoid parasitic reactions without compromising their CO₂ binding ability. Meanwhile, the work also found that the hydrogen atom (H_b) on the lactam-N of isoindigos can induce intermolecular hydrogen bonding with the complexed CO₂ molecule to thermodynamically stabilise the CO₂ adduct. With coupled experimental and computational studies, the authors provide an in-depth analysis of the structure-function relationships of isoindigos as EMCC sorbents. The work is well written and interesting and have significant application prospects in the adsorption and separation of carbon dioxide.

Minor question

1. What are EWGs? The abbreviation needs to be defined before use.
2. Is there any more way to confirm the existence of the isoindigo-CO₂ adduct?

Response to Reviewers' comments on Manuscript "Redox-Tunable Isoindigos for Electrochemically Mediated Carbon Capture" (reference number: NCOMMS-23-18315)

We would like to thank the four reviewers for their positive endorsement of our manuscripts and thoughtful suggestions. All the questions have been answered point-by-point in this response letter. New figures and discussions introduced to the manuscript are highlighted in yellow.

Reviewer #1:

Noteworthy Results of the Paper:

Quinones are the most studied class of molecule for redox-active sorbent based electrochemical CO₂ capture systems however they are limited by their reactivity towards O₂. It has been shown that quinones are limited by a linear free energy relationship which shows that quinones with redox potentials positive of O₂/O₂⁻ do not have a high enough CO₂ binding constant for effective capture from either flue gases or for direct air capture.

In this exciting paper the authors tackle this issue by expanding on previous work by Barlow and Yang (JACS 144, 31, 2022, 14161-14169) (which showed that hydrogen bonding additives stabilise quinone dianions). Here the authors design and synthesise a new type of redox-active sorbent (isoindigos) which are proposed to have strong CO₂ binding due to stabilisation of the CO₂-bound product by H-bonding. This kind of approach goes beyond the work by Barlow and Yang, and avoids the need for volatile additives in a CO₂ capture system. A lot of work has been completed with 21 molecules synthesised and tested.

The central claim of the paper is that the new class of molecules breaks the previously observed scaling relationship, where molecules with more positive potentials bind CO₂ more weakly. If true, this claim would be very interesting. However, I have concerns about the validity of this claim (see below), due to issues with the analysis.

The authors also use some of their isoindigos in a redox-flow setup where they used potentials positive of O₂ reduction and directly measured CO₂ capture and release, demonstrating at least one of their molecules work in ECCS with the presence of O₂ at the bench-top scale, which is exciting.

Are the claims of the study supported?

I see some major issues in the analysis that make me concerned that the claims may not be supported by the data.

(1) A key part of the paper is the measurement of CO₂ binding constants for the various isoindigos. I believe the method used by the authors is incorrect, because the peak shift method cannot be used in cases where the two reduction peaks have merged (sometimes known as the "strong binding" regime). In such cases the maximum observable peak shift in the cyclic voltammogram is determined solely by the separation of the two redox peaks under nitrogen. Use of the peak shift

equation in these cases will therefore always give very similar binding constants for several molecules (since the calculated binding constants now depends solely on the peak separations under nitrogen, which are generally similar for the series of molecules). Indeed we see in Table 1 that many of the molecules are predicted to have very similar identical binding constants (all around 9), which is a key claim of the paper.

This criticism is supported by inspection of the authors' data at 20% CO₂ loading in Figure S3. Comparing 55DBIId and 5BIId in Figure S3 (20% CO₂ data), the smaller peak shift of the two molecules is for 55DBIId, suggesting weaker binding. However, the authors calculate and report that 55DBIId has a slightly higher binding constant than 5BIId. As another example, 55DBIId and IId are reported to have very similar binding constants in Table 1, yet their CVs have very different peak shifts under 20% CO₂. All this underscores the point that the method used to calculate the binding constants has been applied incorrectly.

The above points are problematic for the central claims of the paper, which is that the scaling relationship between redox potential and CO₂ binding constant has been broken by the new class of molecules. A key finding of the work is therefore not supported by the data.

I believe a better method is therefore required to obtain the binding constants. E.g. CO₂ concentrations could be varied to lower values to measure reliable peak shifts, and a linear fit of the peak shift equation could be performed at various concentrations. I think if the authors can update their analyses and claims, the quality of work could be improved significantly.

Response: We appreciate the Reviewer's thoughtful comments on the calculation of the CO₂ binding constant.

We observed through experiments that at relatively low CO₂ concentrations (20% or 10%), the cyclic voltammetry (CV) curves of isoindigos usually exhibit a positively shifted second reduction peak compared to that under N₂; however, most do not completely emerge into the first reduction peak (Supplementary Fig. 3 and 4). **In addition to thermodynamics, this phenomenon can be complicated by kinetic factors.** Namely, during CV scan, there is a kinetic competition [*ACS Sustain. Chem. Eng.* **11**, 11333-11341 (2023)] between the chemical transformation of IId^{•-} into [IId-CO₂]^{•-} ($r = k_{bimolecular}[CO_2][IId^{•-}]$, where [CO₂] and [IId^{•-}] are the concentrations of CO₂ and IId^{•-}) and the electrochemical reduction of [IId-CO₂]^{•-}. Higher CO₂ concentrations lead to higher r , which facilitates the formation of [IId-CO₂]^{•-} and ultimately leads to a single, merged reduction peak. Therefore, the separation between the two cathodic peaks under low CO₂ concentrations can be impacted by CO₂ complexation kinetics. This claim is also supported by our CV studies at different scan rates (**Supplementary Fig. 20**, shown below). Using 6MCIId as an example (the molecule used for EMCC prototype evaluation), it exhibits two reduction peaks under 10% CO₂ at a scan rate of -50 mV s^{-1} . However, the second reduction peak merges into the first as the scan rate decreases, clearly indicating the kinetic effect on the CV behaviour.

Supplementary Fig. 20 | CV of 6MCIId under different atmospheres or at different scan rates. **a**, CV curves under N₂ or 10% CO₂ at a cathodic scan rate of -5 mV s^{-1} . **b**, CV curves under 10% CO₂ at various scan rates from -5 to -200 mV s^{-1} . The CV curves were recorded using 2.5 mM 6MCIId in anhydrous DMF with 0.1 M NBu₄PF₆.

According to the Reviewer's suggestion, we also performed CV measurements of isoindigos at different CO₂ concentrations (**Response Fig. 1**). The relationship between $\Delta E_{\text{peak}(2)}$ and CO₂ concentration is fitted according to the following equation to determine the CO₂ binding constant:

$$\Delta E_{\text{peak}(2)} = \frac{RT}{zF} \ln(1 + K[\text{CO}_2]^n)$$

where F is the Faraday constant, R is the ideal gas constant, T is the temperature, $\Delta E_{\text{peak}(2)}$ is the difference between the second reduction peak potentials (IId^{•-}/IId²⁻) in CO₂ and N₂ atmospheres, $[\text{CO}_2]$ is the concentration of dissolved CO₂ (0.198 M in DMF under 100% CO₂ headspace at 298 K), n is the number of binding sites, and K is the CO₂ binding constant.

As shown in the **Response Fig. 2**, curve fitting gives rise to similar K values on the order of 10^{12} for IId and 6MCIId. However, the fitting quality was unsatisfactory (relatively low R^2 values), probably caused by the abovementioned kinetic effect that leads to an underestimation of $\Delta E_{\text{peak}(2)}$ at low CO₂ concentrations.

We are aware that the current electrochemical method has limitations in calculating strong CO₂-binding molecules, where the binding constant may be underestimated by the CV peak separations under nitrogen. Nonetheless, this method from Bell's seminal work [*SAE Transactions* **97**, 544-552 (1988)] has been widely adopted to derive the binding constants of quinones even for the strong-binding species such as 2,6-ditertbutyl-1,4-benzoquinone [1. Chapter 4 - ELECTROCHEMICAL CONCENTRATION OF CARBON DIOXIDE. In: Sullivan BP (ed). *Electrochemical and Electrocatalytic Reactions of Carbon Dioxide*. Elsevier: Amsterdam, 1993, pp 94-117; 2. *J. Electrochem. Soc.* **150**, D91 (2003); 3. *J. Phys. Chem. C* **126**, 1389-1399 (2022)]. In addition, according to our previous density functional theory (DFT) calculations on strong-

binding molecules, although the potential for second electron transfer can be more positive than the first under CO₂, the difference is usually quite small (< 100 mV) [*Nat. Energy* **7**, 1065-1075 (2022)]. Therefore, as the first study of isoindigos as redox-active CO₂ carriers, we believe the estimation method used in this manuscript can provide a fair comparison with the widely studied quinone molecules.

Response Fig. 1 | CV of IId and 6MCIId under different concentrations of CO₂. **a**, IId. **b**, 6MCIId. The CV curves were recorded using 2.5 mM compound in anhydrous DMF with 0.1 M NBu₄PF₆ at a scan rate of -50 mV s⁻¹.

Response Fig. 2 | Shifts in second reduction peak potentials ($\Delta E_{\text{peak}(2)}$) in the presence of various concentrations of CO_2 calculated using data from Response Fig. 1, and corresponding fitted values of n and CO_2 binding constant K . **a**, IId. **b**, 6MCIId.

In addition, as shown in **Supplementary Fig. 8c** and **d**, the N,N -substituted NNDEHIId and NNDPr66DBIId exhibited two reduction peaks even under 100% CO_2 , suggesting that the intramolecular hydrogen bonding strategy is vital to enhance the CO_2 binding strength.

Supplementary Fig. 8 | CVs of isoindigos with various chemical modifications under different atmospheres. **a**, 5N6MCIId. **b**, 66DMCIId. **c**, NNDEHIId. **d**, NNDPr66DBIId. **e**, 6CIId. **f**, 6CIIdNa. **g**, 6AIIdGly. **h**, 6AIIdSer. **i**, 6B6AIIdSer. The CV was recorded using 2.5 mM compound in anhydrous DMF with 0.1 M NBu₄PF₆ saturated by N₂ (grey), 20% CO₂ (balanced with N₂, blue), and CO₂ (red), respectively, at a scan rate of -50 mV s^{-1} at 298 K.

To further validate the importance of intramolecular hydrogen bonding for CO₂ capture, we have evaluated the performance of NNDEHIId in our flow-based EMCC prototype (**Response Fig. 3**). NNDEHIId, where the amide hydrogens are replaced by 2-ethylhexyl substitute group, is unable to form the intramolecular hydrogen bonding in the CO₂-adduct. The first cycle exhibits a total capture of 0.197 mmol of CO₂ with 0.373 mmol electron input, corresponding to a CO₂ capacity utilization of 53%, which is much lower compared to those isoindigos capable of forming intramolecular hydrogen bonding (up to 80% utilization efficiency). More importantly, NNDEHIId exhibits a sluggish CO₂ release during oxidation and poor cycling performance, decaying substantially in just three cycles.

Response Fig. 3 | Evaluating the performance of NNDEHIId in the flow-based EMCC prototype. CO₂ reading at the exit of the sorbent tank over three repeating capture/release cycles. The liquid sorbent was composed of 12 ml 75 mM NNDEHIId in DMF with 0.25 M NaClO₄ as the supporting salt. The sorbent tank was filled with plastic beads and purged with 10% CO₂ at a flow rate of 2 standard cubic centimetres per minute (sccm). On the opposite side, a ferrocene tank was used to balance the charge, which was filled with 30 ml 0.1 M ferrocene in DMF with 0.25 M NaClO₄ as the supporting salt.

Moreover, we conducted DFT calculations to compare the CO₂ binding energy of isoindigos capable and incapable of forming intramolecular hydrogen bonding. To our delight, the DFT-calculated CO₂ binding energy (E_{b2}) of the two-electron reduced isoindigos agrees well with $\Delta E_{\text{peak}(2)}$ calculated from CV experiments (**Supplementary Table 5**). For isoindigos that can undergo intramolecular hydrogen bonding in their CO₂ adducts, E_{b2} remains relatively constant regardless of EWG substituents. However, E_{b2} substantially decreases in NNDPr66DBIId (incapable of forming hydrogen bonding). These DFT results further corroborate the importance of our structural design by harnessing intramolecular hydrogen bonding to stabilise the CO₂ adduct.

Supplementary Table 5 | DFT-calculated CO₂ binding energies (E_b) of the one-electron and two-electron reduced isoindigos.

	E_{b1} (eV)	E_{b2} (eV)	$\Delta E_{\text{peak}(2)}^A$ (V)
IId	0.08	0.50	0.511
55DMIId	0.08	0.49	0.494
55DBIId	0.20	0.43	0.510
5BIId	0.23	0.50	0.494
66DBIId	0.20	0.45	0.525
NNDPr66DBIId	0.07	0.35	0.247

$^A\Delta E_{\text{peak}(2)}$ was calculated based on the difference between the second reduction peak potentials (IId^{•-}/IId²⁻) in CO₂ and N₂.

(2) I believe there are issues with the interpretation of the NMR spectra in Figure 1b. Two resonances are observed in the range 10-11 ppm for ¹H NMR, and two peaks are observed in ¹³C NMR, close to 180 ppm. The authors write: “The NMR spectra exhibit two types of amide hydrogens and carbonate carbons, which is due to the competition between the hydrogen bonding and Li⁺ complexation (Fig. 2c).” In my opinion this is unlikely to be the correct assignment/explanation, since these species would undergo fast chemical exchange in solution, and would most likely not be resolvable. This raises the question of what is actually forming here. It appears there are two different products present. A full assignment of the entire NMR spectrum for ¹H and ¹³C should be given. DFT calculation of chemical shifts from the already developed structural models could be helpful here.

Response: We thank the Reviewer for their high standard on our NMR characterizations. We have performed extra NMR experiments, including 2D ¹H-¹³C Heteronuclear Single Quantum Coherence (HSQC) NMR and variable-temperature (VT) ¹H NMR, and carefully reexamined our analysis and data interpretation.

We agree with the Reviewer that the crude product of bulk electrolysis contains two different compounds, which are likely the rotational isomers of the isoindigo-CO₂ adducts (IId-CO₂). In the reduced form of isoindigo, the C=C bond connecting the two oxindole rings breaks into C-C bond, allowing the free rotation of the two oxindole rings and giving rise to rotational isomerisation (**Response Fig. 4**). At 20 °C, the crude ¹H NMR after bulk electrolysis suggests the two rotational

isomers exhibit a mole ratio of 2:1, as indicated by the ratio of peak integration at 7.18 and 7.04 ppm. The VT NMR experiment shows that the equilibrium in **Response Fig. 4** shifts to the right as temperature increases (**Supplementary Fig. 3**, shown below). The peak ratio recovers when the temperature returns to 30 °C, manifesting that the conversion between the two isomers is reversible, thereby implying the rotational isomerisation process.

Response Fig. 4 | Rotational isomerisation of IId-CO₂ in the crude solution after bulk electrolysis.

The existence of strong hydrogen bonding can substantially decrease the exchange rate with the solvent, making the proton peaks more resolvable [*Nature*, **262**, 363–369 (1976); *Nucleic Acids Research*, **25**, 8 (1997)]. Unlike proton in hydroxyl or amino groups, the amide proton is much less acidic and the exchange rate in the solution should be sufficiently low to resolve its NMR peaks. To verify this, we performed VT ¹H NMR to study the intramolecular hydrogen bonding by heating the **crude product of bulk reduction of isoindigo** under CO₂ from 20 °C to 90 °C. **Supplementary Fig. 3** shows two sharp peaks at 10.69 and 10.33 ppm at 20 °C, corresponding to the amide protons in the proposed rotational isomers of IId-CO₂, which undergo intramolecular hydrogen bonding with the carbonate oxygen. At elevated temperatures, the peaks broaden and shift upfield, which is consistent with the fact that hydrogen bonding weakens as the temperature increases. At 90 °C, the peaks almost vanish due to the fast proton exchange, suggesting the breaking of the hydrogen bonding. Interestingly, when the temperature is cooled back to 30 °C, the NMR spectrum shows the intact IId-CO₂ adduct. This suggests that our structural design, by harnessing intramolecular hydrogen bonding, can greatly improve the stability of the CO₂ adduct intermediate. Overall, the VT experiment strongly supports the existence of the proposed intramolecular hydrogen bonding between amide hydrogen and carbonate oxygen.

Supplementary Fig. 3 | Variable temperature ^1H NMR spectra of the crude solution from the bulk reduction of IId in CO_2 .

For a full assignment of the NMR spectrum, we conducted 2D HSQC NMR to show the correlation between the carbons and their attached protons. In the 1D spectra of ^1H - ^{13}C HSQC, only proton attached to a carbon can be observed in the ^1H NMR and only carbon attached to protons can be seen in the ^{13}C NMR. It helps us assign the peaks more accurately. For example, the characteristic proton peaks at 10.69 and 10.33 ppm in normal ^1H NMR cannot be observed in HSQC, further suggesting that they are amide protons. In the 2D HSQC spectra, each peak corresponds to a C-H pair, where its two coordinates correspond to the chemical shifts of each carbon and proton atom. The peaks have been assigned to each carbon and proton as shown in **Supplementary Fig. 4** (shown below). Note that the residue peaks at the aliphatic region (~ 4 for ^1H NMR and ~ 46 for ^{13}C NMR) are probably from the electrolytic decomposition of the DMSO solvent. The full assignment of the NMR spectra is given in **Supplementary Fig. 2** (shown below).

Supplementary Fig. 4 | 2D ^1H - ^{13}C Heteronuclear Single Quantum Coherence (HSQC) NMR of the crude solution after bulk reduction of IIId.

Supplementary Fig. 2 | Full assignment of the NMR spectra of IId and IId-CO₂. **a**, ¹H NMR. **b**, ¹³C NMR.

In addition to NMR, we conducted FT-IR measurements of the IId-CO₂ adduct from the crude solution after bulk electrolysis to support the formation of intramolecular hydrogen bonding (**Supplementary Fig. 5**, shown below). Compared to IId, the IR band (1699 cm⁻¹) assigned to C=O stretching vanished, and a new band appeared at 1716 cm⁻¹ in IId-CO₂, supporting the formation of the carbonate bond. Moreover, the IR band of free carboxylate ion should appear at a more red-shifted region (~1600 cm⁻¹) due to the resonance effect. Nevertheless, the new blue-

shifted IR band at 1716 cm^{-1} suggests the formation of carboxylic acid-like species, further confirming the presence of intramolecular hydrogen bonding between the carbonate oxygen and amide hydrogen.

Supplementary Fig. 5 | FT-IR spectra of IId and the IId- CO_2 adduct after bulk electrolysis in DMSO with LiClO_4 as the supporting salt.

Due to the complex conditions, such as the presence of lithium supporting salt and DMSO solvent in the crude product after bulk electrolysis, it is very challenging to build an accurate model to simulate the chemical shift using DFT. However, our newly added 2D HSQC NMR, VT NMR, and FTIR experiments have provided strong evidence for the intramolecular hydrogen bonding in the CO_2 adduct.

(3) Related to the above point, the authors mention that: “Besides, the proton on the lactam N shifts upfield to 10.33 ppm in IId- CO_2 compared to that of 10.89 ppm in the neutral IId, strongly evidencing the formation of intermolecular hydrogen bonding with complexed CO_2 (highlighted in red in Fig. 2a).” I disagree with this statement. Hydrogen bond formation of an N-H group would be expected to cause a shift in a positive direction (due to increased nuclear deshielding), not the reported negative shift change. This again raises the question of what is actually happening here.

This is an important issue to be resolved, since it undermines the central hypothesis that H-bonding is stabilising the bound CO₂ species. Chemical shift calculations with DFT would again be of help here.

Response: We agree with the reviewer that typical hydrogen bonding can lead to deshielding of the proton. However, hydrogen bonding and chemical shift can be affected by many other factors. One explanation of the enhanced shielding can be that the **IId-CO₂ adduct is a two-electron reduced molecule compared to neutral IId**, which increases the global electron density of the molecule and causes the upfield shift of the NMR peak. At elevated temperatures where the hydrogen bonding may break, we do observe the amide proton peaks shift upfield in our VT NMR experiments.

Unlike the intermolecular hydrogen bonding with amino or hydroxyl groups, the intramolecular hydrogen bonding in the IId-CO₂ adduct can be formed in a 6-membered ring between the amide hydrogen and carbonate oxygen at a very small distance (~1.95 Å) supported by our DFT results (**Table 1, Supplementary Figs. 22 and 23**). The lone pair of carbonate oxygen can occupy the anti-sigma bond of amide N-H, partially increasing the electron density of the H and enhancing the shielding. Similar phenomena can be observed in hydrazide species with similar structural motifs. As shown in **Response Fig. 5**, the non-hydrogen-bonded hydrazides exhibit downfield shifted proton NMR peaks (deshielding) compared to hydrogen-bonded ones.

Response Fig. 5 | Examples of intramolecular hydrogen bonding can enhance the shielding effect.

(4) Ideally the ¹H and ¹³C NMR would be shown in Figure 2b before the reaction with CO₂, as well as after, so a comparison can be made. I could not find ¹³C NMR before the CO₂ reaction in the SI, and I think this should be included as a control experiment (sorry if I missed it somewhere!).

The full NMR spectra (the full spectral width) are also not included anywhere as far as I could see, and this should be included so the reader can assess the claims better.

Response: We thank the Reviewer for the suggestion. We have revised **Figure 2** by adding the NMR spectra of IId before bulk electrolysis as a comparison for clarity. The full spectra of the crude solution after bulk electrolysis have been added to the section “*Bulk Electrolysis*” in the Supplementary Information. The ^{13}C NMR spectra of IId have also been added to the Supplementary Information in the section “*Synthesis of Isoindigos*”.

^1H NMR of the crude solution after bulk reduction of IId

¹³C NMR of the crude solution after bulk reduction of IIId

¹³C NMR of IIId

(5) The authors write: “On the contrary, NMR spectra suggest bulk electro-reduction of IIId under CO₂ can yield the proposed adduct with full conversion”. Given the concerns I have about the NMR above, I don’t think this statement is well supported. What is the evidence for “full conversion”?

Response: It shall be noted that the NMR spectra are recorded on the **crude solution** of IId after bulk reduction under CO₂ without further purification. This is highly challenging, and the electrochemically generated CO₂ adduct has not been successfully characterised by NMR for any redox-active CO₂ carriers so far. In contrast, we demonstrate well-resolved, clean NMR spectra of the crude solution, showing that **IId is fully converted (no IId peaks) without the presence of radical species**. Supported by HSQC, VT NMR, and FTIR discussed above, we can conclude with high confidence that the proposed IId-CO₂ adduct structure is the most possible product after bulk electrolysis.

To be more precise, we have modified our statement in the manuscript as below,

[page 6] “On the contrary, crude NMR spectra suggest bulk electro-reduction of IId under CO₂ can yield the proposed adduct with full conversion (**IId was fully consumed**)...”

(6) For the bulk electrolysis tests (Figure 2a,b), can the cumulative charge passed be compared to the theoretical value? This would probe whether or not the reduction of the isoindigo was complete, or partially complete.

Response: As shown in **Response Fig. 6**, there is a dramatic decrease in reduction potential during bulk electrolysis when the capacity reaches 5 mAh (theoretical capacity is 5.36 mAh for 0.1 mmol **IId**), indicating the completion of IId reduction. Our NMR results also show the full consumption of IId by the disappearance of the characteristic proton peak at 10.89 ppm.

Response Fig. 6 | The potential-capacity curve for the bulk electrolytic reduction of IId.

(7) Page 17. The use onset values for calculating theoretical energy efficiencies seem invalid. At the onset potentials a negligible amount of CO₂ capture would take place. So these calculations appear misleading to me.

Response: Indeed, **the theoretical energy was calculated based on the CV peak potentials for CO₂ capture and release**, which is stated in our main text.

[page 18] *“Based on the CV peak potentials for CO₂ capture (reduction) and release (oxidation), we estimated the theoretical minimum energy requirement for CO₂ separation using isoindigo sorbents (Supplementary Fig. 29). Our isoindigo family exhibits minimum energy ranging from 9.8 to 27.6 kJ per mole of CO₂ separated.”*

(8) Can the authors comment on why the second redox process under nitrogen often appears to be less electrochemically reversible (e.g. Figure 1d, and Figure 4a), with a larger peak to peak separation and a more irregular shape?

Response: As shown in **Supplementary Figs. 7 and 8**, among the 21 isoindigos tested, only IId, 6BIId, and 66DBIId exhibit less-defined shapes for the second redox waves in the CV curves. This is probably due to the rotational isomerisation of the one-electron reduced isoindigo radical anions. **Supplementary Fig. 10** (shown below) shows the proposed mechanism of rotational isomerisation. In the radical anion form of isoindigo, the C=C bond connecting the two oxindole rings becomes a single bond, which allows free rotation of the two oxindole rings. This creates various rotational isomers in the one-electron reduced state, giving rise to shoulder peaks in the second redox process. Adding strong electron-withdrawing groups (EWGs) or substitution groups at the 5-position of isoindigo can create dipole moment or steric hindrance, which inhibits the rotational isomerisation of the radical anions and gives rise to a more reversible second redox process. Similarly, adding CO₂ onto isoindigos can also create dipole and steric hindrance, restricting the rotational isomerisation and leading to more defined CV curves.

The following paragraph has also been added to the manuscript.

[page 10] *“Among the 21 examples of isoindigos, only the CV curves of IId, 6BIId, and 66DBIId exhibit less well-defined shapes for the second redox wave. This is probably due to the rotational isomerisation of the one-electron reduced isoindigo radical anions. As shown in Supplementary Fig. 10, the C=C bond connecting the two oxindole rings becomes a single bond in the radical anion form of isoindigo, which allows free rotation of the two oxindole rings. This creates various rotational isomers in the one-electron reduced state, giving rise to shoulder peaks in the second redox process. Adding strong EWGs or substitution groups at the 5-position of isoindigo can create dipole-dipole repulsion or steric hindrance, inhibiting rotational isomerisation and giving rise to a reversible second redox process. Similarly, adding CO₂ onto isoindigos can also create dipole and steric hindrance, impeding the rotational isomerisation and leading to more defined CV curves.”*

Supplementary Fig. 10 | Proposed mechanism for the rotational isomerisation of isoindigo radical anion.

(9). A few of the CVs show extra redox events which aren't discussed in the paper such as for II_d. In the paper they state "Under an inert N₂ atmosphere, II_d exhibits two major redox waves typical to stepwise two-electron transfer within the electrochemical stability window of DMF. Like quinoid species¹³, the two electron transfer steps correspond to the formation of anionic radicals (II_d^{•-}) and dianions (II_d²⁻), respectively" which I find to be somewhat dubious looking at the CV, there should be some discussion of possible extra events.

Response: Please refer to our answer to question 8. The shoulder peaks in CV are likely caused by the rotational isomerisation of II_d^{•-}.

(10) The results from the redox-flow experiments do clearly show however that 6MCIId captures CO₂ upon reduction and releases it upon oxidation (at times however over 100 % of the CO₂ captured is released which at least should have a small amount of discussion). It would have been interesting to see how the performance compared to one of the isoindigos with a CV under N₂ showing less ideal behaviour.

Response: In some cycles, the CO₂ released/captured ratio was calculated to be slightly above 100%. This is limited by the IR-based CO₂ sensor we used for recording the CO₂ level exiting the sorbent tank. Over long-duration continuous recording, the CO₂ baseline of the sensor may drift slightly, giving rise to small margins in estimating the CO₂ release/capture ratios. Nevertheless, the average CO₂ release/capture ratio of all cycles is still slightly lower than unity, indicating the reliability of this analytical method. Moreover, careful baseline correction was implemented during data processing to account for baseline drift in each cycle.

As shown in **Supplementary Fig. 33** (shown below), we have evaluated the EMCC performance of 66DBIId, which possesses shoulder peaks in its CV. 66DBIId achieved an average CO₂ capacity utilisation of 60% and an average release/capture efficiency of >99% over 11 cycles. Interestingly, 87% yield of 66DBIId sorbent can be recovered from the sorbent electrolyte after 11 CO₂ capture-release cycles, corroborating the robustness of the sorbent. In addition, we found ~24% of 66DBIId isomerised into *cis*-66DBIId, supporting our hypothesis on the rotational isomerisation of reduced 66DBIId discussed earlier (**Supplementary Fig. 10**). Nevertheless, the reduction of both 66DBIId and its *cis*-isomer yield the same activated CO₂ sorbent, which we believe does not affect the long-term stability of the EMCC prototype.

Supplementary Fig. 33 | Evaluating the performance of 66DBIId in the flow-based EMCC prototype. **a**, CO₂ reading at the exit of the sorbent tank and voltage curve over repeating capture/release cycles for ~50 h operation. **b**, The CO₂ reading for selected capture/release cycles overlaid, with the cumulative amount of CO₂ captured/released in each cycle relative to the theoretical capacity. Lighter colours represent later cycles. The capture, rest, release, and rest steps are indicated by the shaded regions. For CO₂ capture, 66DBIId was reduced at 10 mA for 60 min followed by a 60 min rest. For CO₂ release, the adducts were oxidised at 10 mA to 0.05 mV followed by a ~120 min constant voltage hold, and finally rested for another 60 min. **c**, The voltage-capacity curve for the 2nd capture/release cycle, indicating an early-stage energy consumption of 166.9 kJ per mol CO₂ concentrated. **d**, The CO₂ capacity utilisation efficiency (purple squares), release/capture efficiency (red crosses), and Coulombic efficiency (grey empty squares) of the system. The liquid sorbent was composed of 12 ml 75 mM 66DBIId slurry in DMAc with 0.25 M NaClO₄ as the supporting salt. On the opposite side, a Fc tank was used to balance the charge, which was filled with 30 ml 0.1 M Fc in DMAc with 0.25 M NaClO₄ as the supporting salt.

(11) The NMR study they performed did show that when the isoindigo-dianion is exposed to CO₂, carbonate peaks appear in the ¹³C NMR, which is evidence that the isoindigos are capturing CO₂ as quinones do, by forming carbonates. In the paper they also state that “Since the discovery of the EMCC mechanism using quinones in 1988, to our best knowledge, the widely accepted quinone-CO₂ carbonate adduct is still a proposed structure and has not been confirmed by non-ambiguous characterisations. This is probably due to the poor stability of the adducts and the transient bonding nature between the reduced sorbent and CO₂” this is not correct, Yang and Barlow (JACS 144, 31, 2022, 14161-14169) have used both NMR and IR previously to show that the dianion of tetrachlorobenzoquinone forms a carbonate when exposed to CO₂.

Response: We thank the Reviewer for pointing this out. In this sentence, we intend to claim that **electrochemically generated** quinone-CO₂ carbonate adduct has yet to be characterized *in-situ* by non-ambiguous methods such as NMR or single crystallography. Based on our experience, bulk reduction of quinone species under CO₂ often yields a mixture of adducts and radical species; the latter interferes with the magnetic field and prevents the collection of good NMR spectra. This may suggest the instability of the quinone-CO₂ adduct, which can be easily oxidized into radicals. In some other cases where the reduced quinones are stable against molecular oxygen, we could not observe a carbonate NMR peak despite their ability to absorb CO₂, and we thus infer that the interaction between these reduced quinones and CO₂ might be relatively weak.

In Yang’s paper, the “chloranil-CO₂ adduct” for NMR characterization is not generated electrochemically. In their method, an excessive amount of methoxide was added to deprotonate the hydroquinone, followed by purging the basic solution with CO₂ to prepare the adduct. Therefore, we believe the adduct structures generated in EMCC are still not well understood.

We have modified our claim in the manuscript for clarification as below.

[page 6] “Since the discovery of the EMCC mechanism using quinones in 1988, to our best knowledge, the widely accepted **electrochemically generated** quinone-CO₂ carbonate adduct is still a proposed structure and has not been confirmed **in an EMCC process** by non-ambiguous characterisations.”

(12) Measurements of the reaction rate of the radical with CO₂ assume an ECE reaction scheme, while the CVs in most cases suggest an EEC mechanism. Is this analysis valid, given this issue?

Response: Given sufficient time for CO₂ binding, the ECEC mechanism is thermodynamically favoured, as supported by our CV experiments at low scan rates (**Supplementary Fig. 20**). Under a constant-potential reduction condition with potential applied at that of the first electron transfer, IId should first be converted into a radical anion and react with CO₂ before the second electron transfer can occur at the applied potential. Therefore, the analysis is valid.

Supplementary Fig. 20 | CV of 6MCIId under different atmospheres or at different scan rates. **a**, CV curves under N_2 or 10% CO_2 at a cathodic scan rate of -5 mV s^{-1} . **b**, CV curves under 10% CO_2 at various scan rates from -5 to -200 mV s^{-1} . The CV curves were recorded using 2.5 mM 6MCIId in anhydrous DMF with 0.1 M NBu_4PF_6 .

Minor comments on the presentation of the work:

Response: We thank the Reviewer for the constructive comments on improving the clarity and presentation of our work.

(13) The inclusion of the biocompatibility work was a bit unexpected. Why is this important? Can some rationale for doing these tests be given for the reader in the main text?

Response: The purpose of the biocompatibility test is to show the low toxicity of our isoindigo-based CO_2 sorbents, underscoring their potential low environmental impact for practical uses. To provide readers with context, the following sentence has been added to the manuscript.

[page 17] “Conventional amine scrubbing sorbents have raised environmental concerns due to their biotoxicity^{30, 31}.”

(14) Figure 1 is a bit hard to interpret for me, even as someone who is familiar with this literature. Is it possible for the authors to simplify it somehow. It seems a strange place to include an equation here.

Response: We have removed the equation in Fig. 1c.

(15) Table 1 caption - please clarify if NMR data are for the molecule before (or after) CO₂ reaction.

Response: The NMR data are recorded using the neutral isoindigo molecules before CO₂ reaction.

We have added the clarification in the caption,

“¹H NMR were recorded on the neutral molecules in DMSO-d₆ using solvent residual peak as the internal reference for calibration.”

Table 1 | The interplay between substituent groups and intramolecular hydrogen bondings in isoindigos and the corresponding impacts on redox potentials (V vs. Fc⁺/Fc) and CO₂ binding.

Isoindigos	¹ H NMR (H ^a) (ppm) ^A	¹ H NMR (H ^b) (ppm) ^A	Bond length of a (Å) ^B	Bond length of b (Å) ^B	Bond length of c (Å) ^B	E _{1/2} (IId/IId ^{•-}) in N ₂ (V vs Fc ⁺ /Fc)	logK[CO ₂]
IId	9.06	10.89	1.963	1.954	1.461	-1.29	9.34
55DMIId	8.85	10.69	1.953	1.970	1.463	-1.29	9.05
5BIId	9.07	10.96	1.962	1.957	1.463	-1.19	9.05
5BIId (EWG) ^C	9.31	11.05	1.936	1.943	1.467	-1.19	9.05
55DBIId	9.32	11.11	1.939	1.946	1.469	-1.09	9.33
6MCIId	9.07	10.94	1.960	1.954	1.464	-1.12	8.89
6MCIId (EWG) ^C	9.15	11.08	1.956	1.944	1.476	-1.12	8.89
66DBIId	8.99	11.10	1.961	1.942	1.468	-1.14	9.59
NNDPr66DBIId	9.05	NA	1.948	NA	1.486	-1.07	4.87

^A¹H NMR were recorded on the neutral molecules in DMSO-d₆ using solvent residual peak as the internal reference for calibration. ^BBond length was obtained from DFT-optimized structures. ^CResults from the non-symmetric oxindole ring with the EWG-substituent.

(16) In the intro, the following sentence could be rewritten to be grammatically correct: “In EMCC, reversible CO₂ capture and release is modulated by applying electrochemical potentials, thereby promising systems that can be operated isothermally at ambient pressure, powered solely by renewable energy sources, and modularly designed to accommodate the multiscale nature of carbon capture needs.”

Response: We break the sentence into two for clarity.

[page 2] *“In EMCC, reversible CO₂ capture and release is modulated by switching electrochemical potentials. Therefore, they can be operated isothermally at ambient pressure, powered by renewable energy sources, and modularly designed to accommodate the multiscale nature of carbon capture needs.”*

(17) Figure 2a. I suggest labelling IId on the figure next to the structure, for clarity.

Response: IId has been labelled next to the structure in Fig. 2a.

(18) Fig. 5a. The connection between the data points with a line is a misleading. The data content in this figure could perhaps be better presented as a table.

Response: The connection has been removed in Fig. 5a. All data content has already been summarised in Supplementary Table 2.

(19) There is a tendency in this paper to have long paragraphs comparing single values against each other which makes it at times quite difficult to follow, take the following sentence, “As an example, anthraquinone exhibits a two-electron-transfer half-wave potential ($E_{1/2}$) of -1.4 V vs. ferrocenium/ferrocene (Fc^+/Fc) in N,N-dimethylformamide (DMF) under CO_2 and a $\log K_{\text{CO}_2}$ of ~ 13.4 . The installation of EWGs, such as one chloro group at 2-position, can anodically shift $E_{1/2}$ to -1.25 V vs. Fc^+/Fc , yet the $\log K_{\text{CO}_2}$ decreased substantially to 2.73 (Supplementary Table 1)²¹.” This information is actually much more clearly shown in Fig. 1, though they don’t reference it.

Response: Fig. 1c has been referenced in this sentence. Moreover, we have thoroughly polished the language of the manuscript to break down long paragraphs and referenced figures for better data visualization wherever possible.

(20) Again with comparing values in text “To verify the hypothesis, we show that EWG substituent at 5-position is more effective in facilitating electro-reduction than that at 6-position. This is because the former is in the ortho-position of H^a and more effective in pulling away the electron density, thereby inducing a stronger hydrogen bonding (Fig. 4d). For instance, 55DBIId and 66DBIId exhibit very close ^1H NMR peaks for H^b (11.11 and 11.10 ppm, Table 1), suggesting the degrees of electron deficiency are very similar in these two molecules. In stark contrast, ^1H NMR spectra suggest a chemical shift of 9.32 ppm for H^a in 55DBIId and 8.99 ppm for that in 66DBIId, clearly indicating a stronger hydrogen bonding (a) in the 5- substituted species. Therefore, $E_{1/2}(\text{IId}/\text{IId}^{\bullet-})$ of 55DBIId (-1.09 V vs. Fc^+/Fc) is more positive compared to 66DBIId (-1.12 V vs. Fc^+/Fc), and this trend is consistent for all the examples in our isoindigo family (Supplementary Table 2, e.g., 5BIId vs. 6BIId and 6MCIId vs. 5NIId vs. 5N6MCIId). This hypothesis is further confirmed with DFT calculations (details vide post).” I find this paragraph very difficult to follow

with having to remember values and then go look at the referenced structures. If this could somehow be turned into a figure the argument I believe would be much clearer.

Response: We thank the Reviewer for the very constructive comment. We have provided a figure as **Supplementary Fig. 12**.

55DBIId

H^a = 9.32 ppm

H^b = 11.11 ppm

$E_{1/2}(\text{IIId}/\text{IIId}^{\bullet-}) = -1.09 \text{ V}$

66DBIId

H^a = 8.99 ppm

H^b = 11.10 ppm

$E_{1/2}(\text{IIId}/\text{IIId}^{\bullet-}) = -1.12 \text{ V}$

Supplementary Fig. 12 | Characteristic ¹H NMR and half-wave redox potential of 55DBIId and 66DBIId.

Reviewer #2:

(1) What are the noteworthy results?

This work presents a new class of molecules, isoindigos, for electrochemically mediated carbon capture (EMCC). Isoindigos have been reported in literature for carbon capture prior to this work as an additive for thermal adsorption-based carbon capture.[1, 2] Isoindigos have also been used as organic semiconductors [manuscript reference 25-28]. This manuscript reports use of isoindigos as organic electrodes for EMCC for the first time. This manuscript does well in presenting isoindigos as a class of molecules for EMCC.

Isoindigos build off observations from the more heavily studied EMCC molecules, quinones, and benefits from the same 1,4-diketone functionality. The biggest benefit of isoindigos over quinones is intramolecular hydrogen bonding from the amide group to the CO₂ adduct. This hydrogen bonding allows for consistent binding potential of isoindigos across the several modified structures evaluated in the manuscript. The hydrogen bonding allows for more flexibility in modifying the isoindigo structure without decrease in CO₂ binding properties, which has been a challenge in other EMCC systems. Modification is desirable to anodically shift the CO₂ binding potential to improve oxidative stability and to improve solubility of the EMCC compound. The several modifications demonstrated in the paper show that this class of molecules are flexible to augmentation which may leave room for further improvement in oxidative stability, solubility, or other desirable features.

[1] Zou, L., et al., Porous Organic Polymers for Post-Combustion Carbon Capture. *Adv Mater*, 2017. 29(37).

[2] Zhao, Y., et al., Isoindigo-based microporous organic polymers for carbon dioxide capture. *RSC Adv.*, 2015. 5(121): p. 100322-100329.

Response: We thank the Reviewer for acknowledging the novelty of our molecular design strategy.

(2) Will the work be of significance to the field and related fields? How does it compare to the established literature? If the work is not original, please provide relevant references.

The significance of this EMCC work is the expansion beyond the more heavily studied EMCC molecules, quinones. Building off quinone knowledge gives credibility to the work. Showing that intramolecular hydrogen bonding in the isoindigo system stabilizes the CO₂ adduct is important for maintaining high CO₂ binding constant ($\log K_{(\text{co}_2)}$) across a range of half potentials ($E_{(1/2)}$). The hydrogen bonding of the isoindigo system is the most interesting and useful benefit of the isoindigos and will likely lead to more research into this new class of EMCC molecules. Other systems have attempted to break the linear $\log K_{(\text{co}_2)}$ vs $E_{(1/2)}$ relationship in a variety of ways but appear to have more difficulties than the isoindigo system. Some modifications have been conducted in the quinone system to incorporate hydrogen bonding, breaking the linear

$\log K_{(\text{CO}_2)}$ vs $E_{(1/2)}$ relationship [manuscript reference 10], but there is less flexibility in designing quinones with hydrogen bonding due to that hydrogen bonding is not being an inherent property of all quinones. Utilization of ionic liquids has aided in hydrogen bonding for quinones, but this may not be ideal for the scaled up EMCC system [manuscript reference 17]. Salts in quinone systems have also been used to shift the potential anodically without reducing binding properties, but like ionic liquids, may not be ideal in the scaled up EMCC system [reference 14]. By using the isoindigo platform, modifications are not required to achieve the stabilizing hydrogen bonding. This leads to greater flexibility in the full system design for future work.

Response: We thank the Reviewer for agreeing with us that our new family of redox-active CO_2 sorbents offers much higher design flexibility than quinones, and the effect of intramolecular hydrogen bonding in isoindigos brings exciting opportunities to break the scaling relationship in sorbent design for practical implementations of EMCC.

(3) Does the work support the conclusions and claims, or is additional evidence needed?

This work does well in supporting the claim that isoindigos break the linear $\log K_{(\text{CO}_2)}$ vs $E_{(1/2)}$ relationship for EMCC capture. This is an important deviation from other EMCC molecule classes which have attempted to use modifications to anodically shift the $E_{(1/2)}$ but have reduced the $\log K_{(\text{CO}_2)}$ such that it may no longer be suitable for carbon capture.

The manuscript uses intentionally poor modifications, replacing the amide hydrogen with an alkane, to highlight that the amide provides stability to the CO_2 adduct. Characterization with H-NMR confirms the hydrogen bonding solidifying the findings in this manuscript. This characterization technique may be of use to other groups who are seeking to add hydrogen bonding stability to their structures. DFT studies support hydrogen bonding playing a role in stabilization of the CO_2 adduct.

Response: We would like to thank the Reviewer for the positive endorsement of our work.

(4) Are there any flaws in the data analysis, interpretation, and conclusions? Do these prohibit publication or require revision?

For the isoindigo molecules presented, the minimum CO_2 capture energy reported as 9.8 to 27.6 kJ mol^{-1} , which is fantastic if it can be realized. The actual cell performance was shown to be 131.5 (isoindigo: 6MCIID, 10mA 60min cycles) to 230.7 kJ mol^{-1} (isoindigo: 6MCIID, 12mA 90min cycles) in 10% CO_2 . The deviation between minimum CO_2 capture energy and actual cell performance appears to be common in quinone papers as well; a low minimum CO_2 capture energy is theorized, but experiments show a higher requirement. Total energy cost appears to be related to the current applied as well as capacity utilization as seen from the above two 6MCIID tests at different currents and capacity utilizations. The authors are encouraged to identify other parameters causing the deviation and suggest possible improvements to achieve capture energy

closer to theory. A few possible parameters are overpotentials due to cell resistance, solubility of CO₂ in the solvent, CO₂ concentration dependence on capture energy, and CO₂ transport.

Response: We strongly agree with the Reviewer that there is a need for device and process engineering to narrow the gap between theoretical and real energy requirements for EMCC. It is noteworthy that the **focus of this work is to present a new class of redox-active CO₂ carriers with robust CO₂ affinity against chemical modification** compared to conventional sorbents such as quinones. The large chemical scope and ease of functionalization allow isoindigos to be customized into different EMCC systems, not limited to the flow-based model in this work.

The flow-based prototype in this work is agreeably not an ultimate design for practical EMCC systems, but only a proof-of-concept demonstration to evaluate the performance of isoindigo sorbents at the lab scale. To improve the energy efficiency of the current system, one big hurdle is the lack of a suitable membrane for non-aqueous electrolytes, which causes large cell resistance and fast crossover of catholyte and anolyte molecules. This issue has also troubled the field of non-aqueous flow batteries for many years [*Chem* **8**, 1611-1636 (2022)]. Furthermore, gas contactors and absorption/desorption columns can be coupled to the system to facilitate CO₂ transport. However, there is a trade-off for the CO₂ solubility, where high solubility is required to facilitate capture from low CO₂ concentration sources, and low solubility is needed to promote release as high CO₂ concentration product streams [*iScience* **25**, 105153 (2022)]. Correspondingly, we have added the following discussion in the main text.

[page 21] *“In this study, we focus on exploring the fundamental chemistry of isoindigos as redox-active CO₂ carriers and their potential to overcome the linear free-energy relationship that limits the structural modification of EMCC sorbents. The flow-based prototype in this work is, however, not an ultimate design for practical EMCC systems but a proof-of-concept demonstration to evaluate the performance of isoindigo at the lab scale. We believe future efforts can substantially improve the EMCC performance by optimizing the electrolytes, electrodes, and membranes of EMCC devices.”*

(5) The experimental energy reported in the flow cell is the first or second cycle. Presumably this is the best cycle and hides variance and degradation. The manuscript does not discuss an average energy cost or standard deviation which may be beneficial.

Response: We thank the Reviewer for this important suggestion. We have now provided the average energy costs with standard deviations for the test using the flow-based EMCC prototype as **Supplementary Fig. 30** (shown below).

Supplementary Fig. 30 | Energy consumption of CO₂ concentration for each cycle in Fig. 6.

We have also added the following discussion to the manuscript,

[page 20] “By integrating the voltage-capacity curve of the first capture-release cycle, the electrical energy consumption under 10% CO₂ is estimated as 127.3 kJ per mol⁻¹ CO₂ in the first cycle and 142.5 ± 8.2 kJ per mol⁻¹ CO₂ over the first 16 cycles (Supplementary Fig. 30), and the value for this early-stage EMCC system is comparable with current carbon capture technologies.”

(6) It would be nice to see a degradation rate which may be possible for the 43 cycles of the 6B6AIIIdSer, but that may not be necessary at this stage.

Response: We thank the Reviewer for providing this critical insight. We have estimated the degradation rate in **Supplementary Note 5**.

Supplementary Note 5 | Degradation rate of EMCC prototype based on 6B6AIIIdSer

To mitigate the crossover problem, the CO₂ adduct was not fully oxidised back to the neutral sorbent in each CO₂ capture-release cycle on purpose, causing a major loss of the electrochemical capacity. Other sources of decay may originate from many different factors, such as crossover of the redox-active electrode molecules, decomposition of the electrolyte or carbon paper, or degradation of the CO₂ carrier. For 6B6AIIIdSer (**Supplementary Fig. 34**), we utilise the energy capacity of the sorbent tank as a reference to estimate the degradation rate using the following equation,

$$r_d = \left(\frac{Q_0 - Q_n}{n \cdot Q_0} \right) * 100\%$$

where r_d is the degradation rate, Q_0 is the initial energy capacity of the sorbent tank, Q_n is the last-cycle energy capacity of the sorbent tank, and n is the cycle number.

Using -1.3 V as the cut-off potential, the energy capacity of the flow cell decreased to 7 mAh in the last cycle tested. The initial energy capacity of 6B6AIIIdSer (0.2 M, 5 ml) is 53.6 mAh.

Therefore, the estimated degradation rate is ~2%. Noticeably, the actual degradation rate should be much less if the CO₂ adduct was fully oxidised in each CO₂ capture-release cycle.

(7) Comparing the 6MCIID test in 10% CO₂ with the 6MCIID in simulated flue gas (10% CO₂, 3% O₂) it is surprising to see a significantly lower capacity utilization (~20%) in the simulated flue gas case. This is despite CV curves under CO₂ and CO₂ + O₂ for 55DBIId being similar (SI Fig 19). Commentary on the deviation may be appropriate.

Response: The CV curves reveal the intrinsic properties of the redox-active CO₂ carriers under ideal conditions (constant CO₂ concentration up to 16% CO₂ and 20% O₂). In our EMCC prototype, the transport of CO₂ across the gas-liquid interface is relatively slow. When the CO₂ concentration drops in the solution, competition between CO₂ complexation and parasitic electrochemical oxygen reduction reactions can occur due to concentration polarizations and hindered adduct formation kinetics. This is a recognized limitation of existing EMCC systems. Nevertheless, developing redox-active CO₂ carriers with reduction potentials more positive than oxygen remains crucial, as it can, at the very least, lessen the parasitic reactions if not completely prevent them.

The parasitic reduction reaction can be further mitigated by process engineering. For example, the CO₂ capture step can be decoupled from the electrochemical reduction step of the sorbents, making the isoindigo-based CO₂ carriers in this work highly desirable.

(8) In the supplementary information, performance of 55DBIId (SI Fig. 24) and 66DBIId (SI Fig. 25) voltage spikes down below the O₂/O₂^{•-} potential in later cycles. This is a concern suggesting degradation of the isoindigo. While a voltage limit can be set as a practical preventative measure to avoid reaching the O₂/O₂^{•-} potential in operation, understanding the degradation method is desirable and could be addressed in the revised manuscript or in future work. A possible degradation path could be side reactions with the IId^{•-} intermediate radical. The radical could pose similar, albeit slower, degradation pathways as the concerning as the O₂/O₂^{•-} pair. At sufficiently high CO₂ concentrations the IId^{•-} may rapidly react with CO₂ with minimal side reactions. At lower CO₂ concentrations there may be more opportunity for side reactions.

Response: Due to the low solubility of 55DBIId and 66DBIId in DMF solvent, we chose DMAc as the solvent when evaluating their EMCC performance, which caused more severe membrane swelling and the consequent crossover problem of redox-active molecules. As a result, the electrochemical capacity of the solution decreases over time, which is the major reason behind the observed voltage spikes.

Using 66DBIId as an example, we utilised NMR to study the degradation of the EMCC prototype. To our delight, the crude ¹H NMR of the sorbent electrolyte after 11 CO₂ capture/release cycles over 50+ hours reveals great stability of 66DBIId (Supplementary Fig. 35, shown below). Nevertheless, the crude NMR spectrum shows the presence of ferrocene in the

sorbent tank at approximately three times the concentration of 66DBIId, indicating the severe crossover issue due to membrane swelling, which led to the decay of the electrochemical capacity.

Furthermore, we can recover the 66DBIId sorbent in 87% yield from the sorbent tank after cycling, suggesting the robustness of the sorbent. In addition, we found ~24% of 66DBIId isomerised into *cis*-66DBIId, supporting our hypothesis on the rotational isomerisation of reduced 66DBIId discussed earlier (See Response to Reviewer 1, question 8). Nevertheless, the reduction of both 66DBIId and its *cis*-isomer can yield the same activated CO₂ sorbent, which we believe does not affect the long-term stability of the EMCC prototype.

At this point, the focus of this work is to fundamentally understand a promising new class of redox-active CO₂ carriers based on isoindigos and interrogate their intrinsic EMCC properties at the molecular level. For the final implementation of this technology, we realize there is a long way to go to optimize the electrodes, electrolytes, membrane separators, device configurations, and many others, which will be our future endeavours as an extension of this work.

Supplementary Fig. 35 | NMR analysis of the sorbent electrolyte after CO₂ capture-release cycling of 66DBIId. The corresponding cycling data are presented in Supplementary Fig. 33. ¹H NMR of 66DBIId (up), crude ¹H NMR of sorbent electrolyte after 11 capture-release cycles (middle), and ¹H NMR of recovered 66DBIId from the sorbent electrolyte.

(9) In general, there is a larger concern for the EMCC field. EMCC technologies have been studied since the seminal paper from W. L. Bell in 1988 [manuscript reference 12]. It is an interesting and promising technology for CO₂ capture and thus has been studied by several groups over the years. Despite this, most references do not actually separate CO₂ from a low concentration stream into a higher stream. Many of the references, including this research, capture and release CO₂ into the same stream. Some references had separate tests in a lower concentrated stream and again a test with higher concentration stream, usually 100% CO₂. This allows for determining a thermodynamic difference between the two states but does not take into consideration transport effects between the two states. One reference conducted a separation effectively from a 100% CO₂ stream into a lower concentration stream [manuscript reference 35]. One reference simulated behavior between 1% and 5% CO₂ [manuscript reference 18]. A lack of full cell operation, fully concentrating from low to high CO₂, makes it difficult to evaluate the full limitations of the technology. There is reference provided that is an exception, Scovazzo, 2003 [reference 16]. Scovazzo, separates CO₂ from 500ppm to 100% using a quinone, DtBBQ. They showed that it was possible experimentally but did not provide an energy cost associated requiring CO₂ to be bubbled through the reduced quinone overnight.

Response: We thank the Reviewer for this very insightful comment and fully agree that there is an urgent need in the EMCC field to thoroughly assess the systems by capturing at low CO₂ feed concentrations and releasing into concentrated streams. We evaluated the electrochemical behaviour of our sorbent using a feed stream of constant concentration, as such configuration enables more reliable quantification of the capture and release capacities of sorbent molecules over continuous cycling without interference from the physical solubility of the electrolyte solvent under different CO₂ headspace. According to the reviewer's suggestion, we tested CO₂ capture under 1% CO₂ + 0.3% O₂ atmosphere (**Supplementary Fig. 37**, shown below) and CO₂ capture/release under 100% CO₂ (**Supplementary Fig. 38**, shown below).

Using 1% CO₂ + 0.3% O₂ as the feed gas, we observe an early-stage energy consumption of 224.2 kJ mol⁻¹ CO₂ captured with >90% removal through a single pass using 6MCIId, as quantified via CO₂ sensor. This suggests that our intramolecular hydrogen bonding strategy effectively improves the CO₂ affinity for low-concentration CO₂ capture.

In another experiment, we demonstrated CO₂ capture-release under 100% CO₂ by using a mass flow metre to record the variation in the exiting gas flow rate. Under a similar condition to that of the low-concentration CO₂ capture, we show an early-stage energy consumption of 143.7 kJ mol⁻¹ CO₂ captured and 13.6 kJ mol⁻¹ CO₂ released, respectively.

We would like to point out that this study focuses on exploring the fundamental chemistry of isoindigos as redox-active CO₂ carriers and their potential to overcome the linear free-energy relationship that limits the structural modification of existing EMCC sorbents. We believe future efforts can substantially improve the EMCC performance by optimizing electrolytes, electrodes, and membranes of the EMCC devices.

Supplementary Fig. 37 | Evaluating CO₂ capture performance of 6MCIId in flow-based EMCC prototype with low-concentration CO₂ feed. 3 sccm 1% CO₂ with 0.3% O₂ was used as the feed gas. **a**, CO₂ reading from an IR-based sensor at the exit of the sorbent tank. **b**, Cumulative amount of CO₂ captured over time. **c**, The voltage-capacity curve of the capture process, indicating an early-stage energy consumption of 224.2 kJ per mole of CO₂ captured. The area of the flow field and the carbon paper electrode was 25 cm². The liquid sorbent was composed of 10 ml 0.25 M 6MCIId in DMF with 0.5 M NaTFSI as the supporting salt. On the opposite side, a Fc tank was used to balance the charge, which was filled with 20 ml 0.25 M Fc in DMF with 0.5 M NaTFSI as the supporting salt and 12.5 mM 6MCIId to mitigate sorbent crossover. We observed faster redox molecule crossover in this system as the membrane area was increased from 5 to 25 cm², resulting in lower CO₂ capacity utilization efficiency.

Supplementary Fig. 38 | Evaluating CO₂ capture-release performance of 6MCIId in flow-based EMCC prototype under 100% CO₂. **a** and **d**, CO₂ flow rate on a mass flow metre at the exit of the sorbent tank. **b** and **e**, Cumulative amount of CO₂ captured and released over time, respectively. **c** and **f**, The voltage-capacity curve of the capture and release process, indicating an early-stage energy consumption of 143.7 kJ per mole of CO₂ captured and 13.6 kJ per mole of CO₂ released, respectively. The area of the flow field and the carbon paper electrode was 25 cm². The liquid sorbent was composed of 10 ml 0.25 M 6MCIId in DMF with 0.5 M NaTFSI as the supporting salt. On the opposite side, a Fc tank was used to balance the charge, which was filled with 20 ml 0.25 M Fc in DMF with 0.5 M NaTFSI as the supporting salt and 12.5 mM 6MCIId to mitigate sorbent crossover.

The following discussion has been added to the main text.

[page 21] “As a final note, we evaluated the CO₂ capture capability of **6MCIId** under low CO₂ concentration and its CO₂ release capability under pure CO₂. Using 1% CO₂ with 0.3% O₂ as the feed, we observe an early-stage energy consumption of 224.2 kJ mol⁻¹ CO₂ and a single pass CO₂ removal of > 90% (Supplementary Fig. 37). This suggests that our intramolecular hydrogen bonding strategy is effective in improving CO₂ affinity for low-concentration CO₂ capture. In another experiment, the CO₂ capture-release behaviour under 100% CO₂ headspace was quantified using mass flow metre (Supplementary Fig. 38). Under a condition similar to that of low-concentration CO₂ capture, we show an early-stage energy consumption of 143.7 kJ mol⁻¹ CO₂ captured and 13.6 kJ mol⁻¹ CO₂ released, respectively.

In this study, we focus on exploring the fundamental chemistry of isoindigos as redox-active CO₂ carriers and their potential to overcome the linear free-energy relationship that limits the structural modification of EMCC sorbents. The flow-based prototype in this work is, however, not an ultimate design for practical EMCC systems but a proof-of-concept demonstration to evaluate the performance of isoindigo at the lab scale. We believe future efforts can substantially improve the EMCC performance by optimizing the electrolytes, electrodes, and membranes of EMCC devices.”

(10) When the end goal of this technology is to separate CO₂ from a low concentration to a high concentration, it should be shown to achieve this. The evidence shown in this manuscript and its whireferences suggest that it is possible to separate CO₂ from a low concentration stream (i.e., 10% flue gas) and concentrate it into higher purity stream but the energies associated are not fully understood. For instance, it is likely that there are inefficiencies due to different amounts of CO₂ dissolved in the solvent in the capture stream versus the release stream as is described in the seminal paper from Bell [manuscript reference 12]. Other inefficiencies are not addressed such as resistance or pH when water is dissolved in the solvent or brought in with the feed stream. This will add to the total energy cost of the EMCC device and assist in determining the viability of the carbon capture method. Furthermore, metrics such as CO₂ flux will need to be considered to assess the cost of materials required for scaleup.

Response: As mentioned above, the emphasis of this study is to examine and evaluate isoindigos as a new class of EMCC sorbents. The ability of isoindigos to capture from a low-concentration stream and release into a high-concentration stream is demonstrated in **Supplementary Fig. 37 and 38**. Nevertheless, further engineering efforts are needed to improve the performance metrics. We agree with the Reviewer that additional inefficiencies will arise due to differences in the physical solubility of CO₂ under different headspace conditions. In particular, the solubility of sorbent must be high enough to overcome such a physical solubility difference [*iScience* **25**, 105153 (2022)]. This also highlights the importance of breaking the scaling relationship in sorbent design such that molecular engineering approaches can be employed to enhance sorbent solubility without compromising CO₂ affinity.

In **Supplementary Fig. 27**, we show 55DBIId is still redox-active and capable of capturing CO₂ in the presence of water (up to 10 vol%). At the current stage, we are aware that solvent resistance in organic systems is a major source of energy penalty in EMCC and limits the CO₂ flux by limiting the practical current density of the devices. Nevertheless, we believe this inefficiency can be addressed with further materials design and engineering efforts [*ChemSusChem* **15**, e202102533 (2022); *ACS EST Engg.* **3**, 1001-1012 (2023)].

Supplementary Fig. 27 | The impact of water on the redox properties and CO₂ binding behaviours of 55DBIId. The CV was recorded using 2.5 mM 55DBIId in DMF with 0.1 M NBu₄PF₆ with various water contents under various atmospheres at a scan rate of -50 mV s^{-1} with an initial cathodic sweeping direction at 298 K.

(11) If these EMCC technologies are being considered for flue gas operation, understanding how the cell performs at sub 1% CO₂ concentrations are important for achieving high removal (>90%) from flue gas streams.[3] Experimental setups in this manuscript and others referenced have not explored sub 1% concentrations with the exception of Scovazzo, 2003 [reference 16]. At lower concentrations, presumably there will be higher tendency for isoindigos or quinones to exhibit both upper and lower reduction peaks which is less prevalent at higher CO₂ concentrations. This may lower the effective capacity of these EMCCs. Furthermore, the second reduction potentials may be below the O₂/O₂^{•-} potential raising concerns for oxidative stability. Thankfully, in this research, there is a compound presented with O₂/O₂^{•-} stability by possessing a second redox potential equal to the O₂/O₂^{•-} potential of -1.35 vs Fc⁺/Fc in DMF (6CIIId SI Fig. 5). All others have a second redox potential below the O₂/O₂^{•-} potential in N₂.

[3] Brandl, P., Beyond 90% capture: Possible, but at what cost? *International Journal of Greenhouse Gas Control*, 2021. 105: p. 103239.

Response: We are grateful for the Reviewer's insight on this issue. The second reduction peak can be tuned by using supporting salts with smaller cations. As shown in **Supplementary Fig. 28**, the second reduction peak can be shifted more positive than the O₂/O₂^{•-} when Na⁺ or Li⁺ salt is used as the supporting electrolyte. An alternative strategy is co-localizing the electrode with CO₂ gas contact by designing gas diffusion electrodes. This will lower the possibility of electrode potential being lowered to the second reduction potential and thermodynamic calculations have indicated the energetic benefit of such design [*Ind. Eng. Chem. Res.* **61**, 10531-10546 (2022)].

Supplementary Fig. 28 | The effect of supporting salts on the redox properties and CO₂ binding behaviours of 6MCIId. a, CV curves of 6MCIId in electrolytes with various supporting salt cations under N₂, 20% CO₂, and CO₂, respectively. **b**, CV curves of 6MCIId in electrolytes with various counter anions under N₂, 20% CO₂, and 100% CO₂. The CV was recorded using 2.5 mM 6MCIId in DMF with 0.1 M supporting salt at a scan rate of -50 mV s^{-1} with an initial cathodic sweeping direction at 298 K. (Grey: N₂; lighter colour: 20% CO₂; colour: 100% CO₂).

(12) This manuscript does mention that for direct air capture (DAC) a binding constant $\log K_{(\text{co}_2)}$ of 5.5 is required to separate from 400 ppm CO₂ (reference 12). This research does not further address DAC, and other referenced EMCC research does not test cycling behavior and energy costs in DAC conditions. Without further evidence, the viability of this technology for DAC is questionable despite its meeting the required $\log K_{(\text{co}_2)}$ binding constant.

Response: We agree with the Reviewer that the viability of this technology for DAC requires further experimental validation. Nevertheless, polymerised anthraquinone has been utilised as the redox-active CO₂ carrier for DAC in a fixed-bed device, with an energy cost from 113 to 900 kJ mol⁻¹ of CO₂ captured depending on the capture rate [*ChemSusChem* **15**, e202102533 (2022)].

With the isoindigo chemistry presented in this work, we provide abundant opportunities for further molecular engineering and process engineering to improve the energy efficiency and reliability of EMCC for DAC applications.

(13) Given that isoindigos are new in the EMCC field, lack of a full energy cost analysis on the system level does not preclude this manuscript from being published. It is hoped that such a full energy cost analysis is forthcoming at some point. In general, this manuscript will help future work to study the different separation use cases expected (Flue gas, DAC, etc.).

Response: We are grateful for the kind suggestion from the Reviewer. To achieve a practical EMCC system, innovations are needed not only at the molecular level but also at the materials and process levels, which cannot be culminated in only one work. We agree that a full energy and techno-economic analysis on the system level is urgently needed to assess the feasibility of EMCC processes. It is indeed our intention for future research.

The vast library of isoindigos in this work will build a foundation for us and others to further push the limit of EMCC. The flow-based EMCC prototype in this work is not the ultimate process design but only a proof-of-concept demonstration to make a fair comparison between our isoindigos and the well-studied quinone species. In the future, we will optimise the system with better electrolytes, membranes, and counter electrode molecules to address the issues of high solvent resistance and membrane crossover. By then, both point-source capture of flue gas and DAC will be tested, and energy cost analysis will become more meaningful on such optimised systems.

(14) There are a couple further questions in the manuscript. First is why biocompatibility is brought up at all in this manuscript? The supplementary information shows that mammalian cell tests were conducted and showed adding -Ser modification allowed for the isoindigo to be more biocompatible than the base isoindigo. Only one sentence was given in the manuscript on biocompatibility along with two words in the conclusion, “good biocompatibility”. Why is this important? Is this supposed to show it is safer? Are there concepts that EMCC would be used in a biological system? It is interesting as a curious test, but it is not clear how the result fits in. No citations are given about the need for biocompatibility nor methodology. Does the 5% CO₂ incubation condition influence the results. Considering that this compound is intended to capture and release CO₂, would a lower or higher CO₂ concentration matter? Would an isoindigo cycling between CO₂ adduct form and CO₂ free form affect the biocompatibility? The authors are suggested to fully explain the purpose for these tests or remove them from the paper. In its current form, the mammalian cell test distracts from the main purpose of the paper which is EMCC.

Response: We apologize for the lack of context on biocompatibility in the manuscript.

A concern for amine scrubbing technologies is the use of toxic amine sorbents, giving rise to potential environmental impacts via fugitive emissions. Our cytotoxicity test suggests our

modified isoindigo is biocompatible, improving its value for practical applications. The 5% CO₂ incubation condition is standard for cell culture and the cytotoxicity of the CO₂ adducts can be an interesting topic for future investigations.

We have added the following sentences to justify the biocompatibility experiments.

[page 17] “Conventional amine scrubbing sorbents have raised environmental concerns due to their cytotoxicity^{30, 31}.”

We have added citations related to the biocompatibility of amines as Ref. 30 and 31 in the main text. Additional references on the methodology have been included as Ref. 9 and 10 in the Supplementary Information.

30. del Rio, B. *et al.* The biogenic amines putrescine and cadaverine show in vitro cytotoxicity at concentrations that can be found in foods. *Sci. Rep.* **9**, 120 (2019).

31. Rohr, A. C. *et al.* Potential toxicological effects of amines used for carbon capture and storage and their degradation products. *Energy Procedia* **37**, 759-768 (2013).

9. Rampersad, S. N. Multiple applications of alamar blue as an indicator of metabolic function and cellular health in cell viability bioassays. *Sensors* **12**, 12347-12360 (2012).

10. Kumar, P., Nagarajan, A. & Uchil, P. D. Analysis of cell viability by the alamarBlue assay. *Cold Spring Harb. Protoc.* **6**, 462-464 (2018).

(15) Is the methodology sound? Does the work meet the expected standards in your field?

Yes, the methodology is built upon related EMCC technologies. The methodologies presented provide parameters that can be used to compare across the field. DFT studies support hydrogen bonding playing a role in stabilization of the CO₂ adduct. H-NMR being used as a tool to verify hydrogen bonding further supports the claims presented. The numerous substituent groups studied in the paper provide experimental examples of how different moieties may affect the isoindigo CO₂ binding properties. This provides a good starting point into future isoindigo EMCC research.

Response: We are grateful for the positive endorsement of our methodology.

(16) Is there enough detail provided in the methods for the work to be reproduced?

Yes, I appreciated the details on the reaction setup and yields for each of the isoindigos presented in the paper. If possible, it would be good to understand if side reactions are present, such as the formation of the indigo isomer. For the flow based EMCC prototypes sufficient operating parameters and design setup are provided to replicate.

Response: We are grateful for the positive endorsement regarding the experimental details provided in our manuscript. We agree with the Reviewer that future studies are needed to interrogate the detailed degradation mechanisms of the EMCC sorbent, and we are currently working on this topic.

(17) Some minor point: EWG appears not defined when first used. It might be a good idea to show the O₂ redox potential in one of the figures to pictorially display which compound has a more positive redox potential.

Response: The definition of “EWG” has been given in the Introduction.

[page 2] “The installation of *electron-withdrawing groups (EWGs)*, such as one chloro group at 2-position,..”

The requested figure showing the redox potentials of isoindigo molecules with respect to oxygen reduction potential is already provided as **Supplementary Fig. 9**.

Supplementary Fig. 9 | Tabulated half-wave potentials of isoindigos under N_2 or CO_2 . Half-wave potentials are summarised from the CVs of various isoindigos using 2.5 mM compound in DMF with 0.1 M NBu_4PF_6 under N_2 (blue) or CO_2 (pink) (filled circle: first redox potential; empty circle: second redox potential). The oxygen reduction reaction to superoxide occurs at -1.35 V vs. Fc^+/Fc in DMF.

Reviewer #3:

Response: We sincerely appreciate the Reviewer's valuable suggestions in improving our manuscript.

Reviewer #4:

The present study designed and prepared a new class of bifunctional redox-tunable CO₂ carriers based on isoindigo and their derivatives. The unique intramolecular hydrogen bonding in isoindigo moieties can be tuned by chemical modification on different sites, which can promote electron reduction, change solubility and avoid parasitic reactions without compromising their CO₂ binding ability. Meanwhile, the work also found that the hydrogen atom (H^b) on the lactam-N of isoindigos can induce intermolecular hydrogen bonding with the complexed CO₂ molecule to thermodynamically stabilise the CO₂ adduct. With coupled experimental and computational studies, the authors provide an in-depth analysis of the structure-function relationships of isoindigos as EMCC sorbents. The work is well written and interesting and have significant application prospects in the adsorption and separation of carbon dioxide.

Response: We sincerely appreciate the positive endorsement of our work from the Reviewer.

Minor question

(1) What is EWGs? The abbreviation needs to be defined before use.

Response: We apologize for not properly defining this abbreviation. EWG stands for electron-withdrawing groups. A definition has been added to the manuscript.

[page 2] “The installation of **electron-withdrawing groups (EWGs)**, such as one chloro group at 2-position,..”

(2) Is there any more way to confirm the existence of the isoindigo-CO₂ adduct?

Response: In addition to ¹H and ¹³C NMR, we performed 2D HSQC NMR, VT-NMR, and FTIR to further confirm the existence of the isoindigo CO₂ adduct. Please also refer to our answer to Question 2 from Reviewer 1.

For example, we conducted FT-IR measurements of the crude isoindigo (**IId**) solution after bulk electrolysis under CO₂ to support the formation of the IId-CO₂ adduct. Compared to **IId**, the IR band (1699 cm⁻¹) assigned to C=O stretching vanished, and a new band appeared at 1716 cm⁻¹ after bulk electrolysis (**Supplementary Fig. 5**, shown below), **supporting the formation of the carbonate bond.**

Supplementary Fig. 5 | FT-IR spectra of IId and the IId-CO₂ adduct after bulk electrolysis in DMSO with LiClO₄ as the supporting salt.

REVIEWER COMMENTS

Reviewer #1 (Remarks to the Author):

Comments on response to reviewer 1:

The authors have done some really nice new science to address the comments - great job! However, lots of the very interesting new data and discussion in the rebuttal letter has not been added to the paper. I suggest some quick final revisions to update the paper to reflect to the exciting new work that was completed, and most importantly to properly represent the CO₂ binding constants which are at the heart of this work.

- review 1, point 1: the authors have done very nice new measurements of the binding constants (response fig. 1 and 2), where they varied the CO₂ concentration and performed a fit to get K values. These new analyses give more reliable measurements of the CO₂ binding constant K, since much more data is used (rather than a single cyclic voltammogram dataset). The new calculated binding constants are more than three orders of magnitude than the ones included in the paper (i.e. the new analysis shows these molecules are better at capturing CO₂ than is currently reported in the manuscript!)

I therefore do not understand why the authors have not updated the manuscript with this new more detailed and accurate analysis. It would be very nice science to add this in, and have some discussion. Please can the authors include the analysis in the paper, so a discussion of the possible errors of log(K) can be discussed (e.g. by comparing the methods)? This would seem to me to be the most transparent and open way to present the results. The best way to do this, in my opinion, would be to present both analysis methods in the paper, and discuss the differences. This would be very helpful for the community.

As the paper stands, it gives the impression that the log(K) values can reliably be determined with the experimental method used - which is misleading in my opinion, especially given the new data the authors have presented.

- also on reviewer 1, point 1:

If kinetics influence peak positions, does this add further uncertainty to the CO₂ binding constant calculation? Can some discussion of this be added to the paper?

Relatedly, what scan rate was used to calculate the binding constants? I could not find this in the Methods sections. Sorry if I missed this.

- Regarding the new data in Supplementary table 5 - can experimental and theoretical CO₂ binding constants also be compared directly as log(K) values?

- In some other cases the authors have been some really nice points and/or shown new data in the response document that they have not included in the revised manuscript nor SI.

E.g. can the data from response figure 6 be added to the manuscript SI? Otherwise readers of the published paper may have the same questions, which would be left unanswered.

E.g. the response to point 10 includes interesting discussion which could be added to the manuscript.

- point 7, the authors have not addressed the comment here. As mentioned, using onset potentials is misleading in theoretical energy efficiency calculations - since almost no CO₂ would be captured at these potentials. Can this method be justified in the text? If not, a better method for estimating theoretical energy efficiencies is likely required.

Reviewer #2 (Remarks to the Author):

Firstly, the effort to improve the manuscript is appreciated. The responses to comments of all the reviewers have improved this manuscript. In general, the information presented is important to publish and expand knowledge in this field.

On page 14, 20% or 10% CO₂ concentrations are not considered relatively low. The first sentence of the paper is proposing this CO₂ capture technology to mitigate climate change. Many flue gases do not get to 20%, and flue gases are a primary point source target for capturing CO₂. 20% CO₂ concentration is a reasonable upper bound, and 10% is a good condition to look at for lower CO₂ concentration flue gases.

The addition of standard deviations and degradation rates for the CO₂ capture tests is appreciated. It gives the reader a more accurate understanding of current performance. Selecting only the best performing curve, hides this from readers.

On page 21, a simple reword may help. In the same sentence negligible voltage decay and degradation rate is discussed. It makes it seem as though a voltage degradation rate is being discussed not capacity degradation rate as listed in the SI. These are not the same.

The test at 1% CO₂ shows continued performance at low concentration and the range of viability for these compounds. This helps suggest a stream concentration delta of 9-19% CO₂ from the inlet to outlet can be maintained given enough of the sorbent present in a system. Future work showing continuous stream processing would be encouraging for the prospects of the technology.

The test at 100% CO₂ shows the energy cost to release the CO₂ into a purified stream. It is not negligible as it is shown to be 13.6 kJ/mol in one cycle. This is on the order of the theoretical thermodynamic energy requirement for these compounds. It shows that the release portion of this system will have an energy component to it, but it is low which is encouraging for the technology going forward. From the references in the manuscript, it appears as though this test and energy cost associated has not been shown before with quinones in a single cycle.

The additional discussion on page 21 differentiating the intended focus of the paper, isoindigo characterization, and the use case of carbon capture helps the reader see that the EMCC sorbents are just one part of a complex system. It sounds like someone in the community should make engineering and research efforts in other parts of the system with the electrolyte, electrodes, and membranes. These may be large hurdles given the crossover issues discussed in the manuscript and should be considered for future work.

On Biototoxicity/biocompatibility-----

Despite additional commentary on biotoxicity and biocompatibility it does not appear to be sufficient to claim "good biocompatibility". The discussion has been expanded to one paragraph listing the LC₅₀ value for IId and 6AIIdSer. This is an internal comparison between two isoindigos and background with DMF which suggests improved with the Ser modification.

It is suggested that amines have environmental biotoxicity concerns, but the discussion does not directly compare to other amines. In the added reference from Rohr 2013 (Reference 32) a suggestion that MEA, used in carbon capture, may cause breathing irritation. In the added reference from del Rio 2019 (Reference 31) IC₅₀ levels are given for putrescine at 40 mM, where as mM 6AIIdSer LC₅₀ is 0.2 mM with appropriate conversion. The IC₅₀ and LC₅₀ are not directly comparable making this a tenuous discussion. It is not even clear if putrescine is a relevant compound for carbon capture as the IC₅₀ is from a study on toxicity in food packaging.

The isoindigo biocompatibility is not compared to other EMCC sorbents, yet this is listed as one of the 4 main benefit in the conclusion.

In an SDS for MEA from Sigma Aldrich Section 12.1 LC₅₀ for *Cyprinus caprio* (Carp)-349ug/ml

In an SDS for putrescine from Sigma Aldrich Section 11.1 LC₅₀ inhalation rat-1.131ug/ml; Section 12.1 LC₅₀ *Poecilia reticulata* (guppy)-730ug/ml

In an SDS on isoindigo from CaymanChem there are no currently known LC₅₀s

Given that no isoindigo biotoxicity tests have been given as reference, perhaps this data is still valuable to publish. It appears as though the data is an interesting cytotoxicity test. It does not appear as though these compounds have definitively been shown to be biocompatible. It is encouraged to not list this as a major advantage in the conclusion. Instead, biotoxicity has been explored with a cytotoxicity test. It may

be promising for future work, but it is hard to argue “good biocompatibility” has been concluded with the present data.

Very minor comment. Fig. 3 R1 and R2 are listed. R1 is the N-substitution modification but it is labeled as simply R and not R1.

Reviewer #3 (Remarks to the Author):

Reviewer #4 (Remarks to the Author):

I would like to recommend this article to be published in Nature Communications

Response to Reviewers' comments on Manuscript "Redox-Tunable Isoindigos for Electrochemically Mediated Carbon Capture" (reference number: NCOMMS-23-18315A)

We would like to thank the four reviewers for their positive endorsement of our manuscripts and thoughtful suggestions. All the questions have been answered point-by-point in this response letter. New figures and discussions introduced to the manuscript are highlighted in yellow.

Reviewer #1:

The authors have done some really nice new science to address the comments - great job!

Response: We are grateful for the positive endorsement from this reviewer.

However, lots of the very interesting new data and discussion in the rebuttal letter has not been added to the paper. I suggest some quick final revisions to update the paper to reflect to the exciting new work that was completed, and most importantly to properly represent the CO₂ binding constants which are at the heart of this work.

Response: The required information has been added to the manuscript.

(1) review 1, point 1: the authors have done very nice new measurements of the binding constants (response fig. 1 and 2), where they varied the CO₂ concentration and performed a fit to get K values. These new analyses give more reliable measurements of the CO₂ binding constant K, since much more data is used (rather than a single cyclic voltammogram dataset). The new calculated binding constants are more than three orders of magnitude than the ones included in the paper (i.e. the new analysis shows these molecules are better at capturing CO₂ than is currently reported in the manuscript!)

I therefore do not understand why the authors have not updated the manuscript with this new more detailed and accurate analysis. It would be very nice science to add this in, and have some discussion. Please can the authors include the analysis in the paper, so a discussion of the possible errors of log(K) can be discussed (e.g. by comparing the methods)? This would seem to me to be the most transparent and open way to present the results. The best way to do this, in my opinion, would be to present both analysis methods in the paper, and discuss the differences. This would be very helpful for the community.

Response: We express our sincere gratitude for the recommendations provided. Our initial decision not to include the new analysis was driven by an intention to prevent potential confusion among readers due to the presentation of two distinct sets of binding constants, each derived through different methodologies. Besides, the fitting was unsatisfactory due to the low R^2 value and unreasonable binding sites ($n \gg 2$). Heeding the reviewer's advice, we have now incorporated this analysis into Supplementary Fig. 39 and 40 in the revised manuscript.

Supplementary Fig. 39 | CV of IId and 6MCIId under different concentrations of CO₂. **a**, IId. **b**, 6MCIId. The CV curves were recorded using 2.5 mM compound in anhydrous DMF with 0.1 M NBu₄PF₆ at a scan rate of -50 mV s^{-1} .

Supplementary Fig. 40 | Shifts in second reduction peak potentials ($\Delta E_{\text{peak}}(2)$) in the presence of various concentrations of CO_2 calculated using data from Response Fig. 1, and corresponding fitted values of n and CO_2 binding constant K . **a**, IId. **b**, 6MCIId.

A short discussion has also been added to the manuscript.

[page 21] **“Comparison of methods for estimating CO_2 binding constants**

As a final note, in this manuscript, we estimated the K_{CO_2} of isoindigos using the prevalent method adopted for quinones, bipyridines, and benzyl thiolate^{12, 21, 22}, providing an equitable comparison with previously reported redox-active CO_2 carriers. Accordingly, we assume that the two-electron-reduced isoindigo binds to one molecule of CO_2 and calculate the K_{CO_2} based on $\Delta E_{\text{peak}}(2)$ under pure N_2 and CO_2 atmosphere, respectively (Supplementary Note 3). This method, therefore, eliminates the kinetic effects in CV under lower CO_2 concentrations. Alternatively, we recorded the CV of **IId** and **6MCIId** under various CO_2 concentrations (Supplementary Fig. 39) and fitted the relationship between $\Delta E_{\text{peak}}(2)$ and CO_2 concentration (Supplementary Fig. 40 and Supplementary Note 7). Similar K_{CO_2} values were obtained for **IId** and **6MCIId** on the order of 10^{12} , further confirming the high tolerance of EWG in isoindigo structural motifs for strong CO_2 binding. However, the fitting quality was unsatisfactory, with low R^2 values and unreasonable number of binding sites. Therefore, we took the former method to calculate K_{CO_2} for all isoindigos reported in this work.”

The details of the fitting have been added as Supplementary Note 6.

“Supplementary Note 6 | An alternative method to estimate the CO_2 binding constant

As an alternative method, we also performed CV measurements of isoindigos at different CO_2 concentrations (Supplementary Fig. 39). The relationship between $\Delta E_{\text{peak}}(2)$ and CO_2 concentration is fitted according to the following equation to determine the CO_2 binding constant:

$$\Delta E_{\text{peak}}(2) = \frac{RT}{zF} \ln (1 + K[\text{CO}_2]^n) \quad (\text{Eq 10})$$

where F is the Faraday constant, R is the ideal gas constant, T is the temperature, $\Delta E_{\text{peak}}(2)$ is the difference between the second reduction peak potentials ($\text{IIId}^{\bullet-}/\text{IIId}^{2-}$) in CO_2 and N_2 atmospheres, $[\text{CO}_2]$ is the concentration of dissolved CO_2 (0.198 M in DMF under 100% CO_2 headspace at 298 K), n is the number of binding sites, and K is the CO_2 binding constant.

As shown in **Supplementary Fig. 40**, curve fitting gives rise to similar K values on the order of 10^{12} for IIId and 6MCIId . However, the fitting quality was unsatisfactory (relatively low R^2 values and unreasonable binding sites >2), likely due to the limited CO_2 complexation kinetics under low CO_2 concentrations that complicated the thermodynamic analysis.”

(2) As the paper stands, it gives the impression that the $\log(K)$ values can reliably be determined with the experimental method used - which is misleading in my opinion, especially given the new data the authors have presented.

Response: We would like to clarify our reason for using the original method to compare $\log(K)$ values among different redox-active CO_2 carriers. The method is the most prevalently adopted approach in the field and has been employed to estimate the CO_2 binding constants of redox-active CO_2 carriers such as reduced quinones, bipyridine, and benzyl thiolate [*SAE Transactions* **97**, 544-552 (1988); Chapter 4 - ELECTROCHEMICAL CONCENTRATION OF CARBON DIOXIDE. In: Sullivan BP (ed). *Electrochemical and Electrocatalytic Reactions of Carbon Dioxide*. Elsevier: Amsterdam, 1993, pp 94-117; *J. Am. Chem. Soc.* **144**, 14161-14169 (2022); *J. Phys. Chem. C* **126**, 1389-1399 (2022).] Given that the focus of this manuscript is to present a new class of redox-active sorbents based on isoindigos, we believe that adhering to a consistent $\log(K)$ estimation methodology will facilitate fair comparison with previously reported CO_2 carriers even if the binding constants are increased by three orders using the new analytical method.

Further, we agree with the reviewer that new analytical methods should be developed in the future to enhance the accuracy in calculating the binding constants of redox-active CO_2 carriers.

(3) also on reviewer 1, point 1: If kinetics influence peak positions, does this add further uncertainty to the CO_2 binding constant calculation? Can some discussion of this be added to the paper?

Response: We thank the reviewer for pointing this out. The current calculation method only takes into account the cyclic voltametric behavior of the molecules under pure N_2 and CO_2 . The CV peaks in these two conditions shift negligibly under different scan rates, indicating the reaction has reached an equilibrium state and kinetics does not influence the peak positions in the pure gas conditions.

The discussion has been included in the section “**Comparison of methods for estimating CO₂ binding constants**”.

(4) Relatedly, what scan rate was used to calculate the binding constants? I could not find this in the Methods sections. Sorry if I missed this.

Response: As discussed above, the scan rate does not affect the calculation of the binding constants. The scan rate has been labeled in the captions Fig. 2 and 4 and Supplementary Fig. 7, 8, 20 and 39. All the CV experiments adopt a scan rate of -50 mV s^{-1} except for Supplementary Fig. 20 and 39.

(5) Regarding the new data in Supplementary table 5 - can experimental and theoretical CO₂ binding constants also be compared directly as log(K) values?

Response: Due to the limitation of DFT modeling, we have to assume an ECEC process and calculate the CO₂ binding energies of IId^{•-} (E_{b1}) and [IId-CO₂]^{•-} (E_{b2}) separately.

The theoretical CO₂ binding constant ($\log K_{\text{CO}_2}$) was calculated by the equation below:

$$\log K_{\text{CO}_2} = \frac{E_{b1} + E_{b2}}{RT \ln 10}$$

where R is the ideal gas constant and T is the room temperature of 298 K.

As shown in the updated Supplementary Table 5, the DFT calculation is consistent with the trend of our experimental results. NNDPr66FBIId, which is unable to form intramolecular hydrogen bonding in the CO₂ adduct, exhibits a dramatic decrease of $\log K_{\text{CO}_2}$ compared to non- N -substituted isoindigos. This further validates our structural design to break the linear free energy relationship in redox-active CO₂ carriers.

Accordingly, we have updated Supplementary Table 5. We have also revised the description in the manuscript as below,

[Page 15] “Besides, DFT-calculated CO₂ binding constants ($\log K_{\text{CO}_2}$) of the isoindigos agree well with the trend of our experimental results (Supplementary Table 5).”

Supplementary Table 5 | DFT-calculated CO₂ binding energies of IId^{•-} (E_{b1}) and [IId-CO₂]^{•-} (E_{b2}).

	E_{b1} (eV)	E_{b2} (eV)	Theoretical $\log K_{\text{CO}_2}$ ^A	Experimental $\log K_{\text{CO}_2}$
IId	0.08	0.50	9.18	9.34
55DMIId	0.08	0.49	9.03	9.05

55DBIId	0.20	0.43	9.98	9.33
5BIIId	0.23	0.50	11.56	9.05
66DBIId	0.20	0.45	10.29	9.59
NNDPPr66DBIId	0.07	0.35	6.65	4.48

^ATheoretical $\log K_{\text{CO}_2}$ was calculated using the equation, $\log K_{\text{CO}_2} = \frac{E_{\text{b}1} + E_{\text{b}2}}{RT \ln 10}$, where R is the ideal gas constant and T is the room temperature of 298 K.

(6) In some other cases the authors have been some really nice points and/or shown new data in the response document that they have not included in the revised manuscript nor SI. E.g. can the data from response figure 6 be added to the manuscript SI? Otherwise readers of the published paper may have the same questions, which would be left unanswered. E.g. the response to point 10 includes interesting discussion which could be added to the manuscript.

Response: The Response Figure 6 has been added to the Bulk Electrolysis section on Page 32 in the Supplementary Information.

(7) point 7, the authors have not addressed the comment here. As mentioned, using onset potentials is misleading in theoretical energy efficiency calculations - since almost no CO_2 would be captured at these potentials. Can this method be justified in the text? If not, a better method for estimating theoretical energy efficiencies is likely required.

Response: We are afraid that there is a misunderstanding of our analytical methods by the reviewer. In this manuscript, we **never** report the **onset potentials** and use them to estimate the theoretical energy efficiency. The values reported in Table 1, Supplementary Table 2 and 5, and Supplementary Fig. 29 are all based on **the peak potentials** in the CV curves, where the reduction/oxidation currents reach global maxima. This has also been stated in our manuscript.

[page 18] “Based on the **CV peak potentials** for CO_2 capture (reduction) and release (oxidation), we estimated the theoretical minimum energy requirement for CO_2 separation using isoindigo sorbents (Supplementary Fig. 29). Our isoindigo family exhibits minimum energy ranging from 9.8 to 27.6 kJ per mole of CO_2 separated.”

It is noteworthy that our isoindigo sorbents have smaller energy gaps between CO_2 capture and release compared to those of strong-binding quinones [*Joule* **6**, 221-239 (2021)] and sp^2 -N Lewis bases [*Nat. Energy* **7**, 1065-1075 (2022)], which is highly desired for constructing efficient EMCC systems.

Reviewer #2:

Firstly, the effort to improve the manuscript is appreciated. The responses to comments of all the reviewers have improved this manuscript. In general, the information presented is important to publish and expand knowledge in this field.

Response: We appreciate the positive endorsement from this reviewer.

(1) On page 14, 20% or 10% CO₂ concentrations are not considered relatively low. The first sentence of the paper is proposing this CO₂ capture technology to mitigate climate change. Many flue gases do not get to 20%, and flue gases are a primary point source target for capturing CO₂. 20% CO₂ concentration is a reasonable upper bound, and 10% is a good condition to look at for lower CO₂ concentration flue gases.

Response: We thank the reviewer for the great suggestions. We have modified the corresponding sentence as below,

[page 14] “*At a CO₂ concentration of 20% or 10%, the CV curves of isoindigos usually exhibit a positively shifted second reduction peak compared to that under N₂; however, most do not completely emerge into the first (Supplementary Fig. 7 and 8).*”

(2) The addition of standard deviations and degradation rates for the CO₂ capture tests is appreciated. It gives the reader a more accurate understanding of current performance. Selecting only the best performing curve, hides this from readers.

On page 21, a simple reword may help. In the same sentence negligible voltage decay and degradation rate is discussed. It makes it seem as though a voltage degradation rate is being discussed not capacity degradation rate as listed in the SI. These are not the same.

Response: We thank the reviewer for the very constructive comment. We have reworded the corresponding sentence as below,

[page 21] “*Moreover, **6B6AIIIdSer** shows excellent cycling stability with negligible voltage decay over >40 cycles and ~200 hours of operation with a **capacity** degradation rate of 2% (Supplementary Note 5).*”

(3) The test at 1% CO₂ shows continued performance at low concentration and the range of viability for these compounds. This helps suggest a stream concentration delta of 9-19% CO₂ from the inlet to outlet can be maintained given enough of the sorbent present in a system. Future work showing continuous stream processing would be encouraging for the prospects of the technology.

Response: Thank you for the suggestion. With the promising isoindigo sorbents in hand, we believe continuous steam processing can be realised by prospective process engineering in the near future.

(4) The test at 100% CO₂ shows the energy cost to release the CO₂ into a purified stream. It is not negligible as it is shown to be 13.6 kJ/mol in one cycle. This is on the order of the theoretical thermodynamic energy requirement for these compounds. It shows that the release portion of this system will have an energy component to it, but it is low which is encouraging for the technology going forward. From the references in the manuscript, it appears as though this test and energy cost associated has not been shown before with quinones in a single cycle.

Response: The primary aim of this paper is to present a new class of redox-active CO₂ carriers that overcomes the linear-free-energy-relationship constraints in conventional sorbents such as quinones. We are optimistic that subsequent advances in process engineering will lower this energy cost by optimising the electrolytes, electrodes, and membranes of the EMCC devices.

We express our sincere gratitude for the insightful suggestions and stringent criteria provided by the reviewer. In alignment with your suggestions, we have taken the initiative to be the pioneers in undertaking these specific tests and energy assessments in this field. We anticipate that our findings will substantially contribute to the ongoing evolution of this technology, serving as pivotal references for future explorations and applications.

(5) The additional discussion on page 21 differentiating the intended focus of the paper, isoindigo characterization, and the use case of carbon capture helps the reader see that the EMCC sorbents are just one part of a complex system. It sounds like someone in the community should make engineering and research efforts in other parts of the system with the electrolyte, electrodes, and membranes. These may be large hurdles given the crossover issues discussed in the manuscript and should be considered for future work.

Response: Indeed, the role of the ferrocene molecule is just a counter electrolyte to balance the charge of the EMCC system. There is a huge chemical space of redox-active organic materials (ROM), which can be tested to further optimise the efficiency of the system. One way to circumvent the crossover issues is to build symmetric cells, which is a direction we are currently working on.

(6) On Biototoxicity/biocompatibility

Despite additional commentary on biotoxicity and biocompatibility it does not appear to be sufficient to claim “good biocompatibility”. The discussion has been expanded to one paragraph listing the LC50 value for IId and 6AIIdSer. This is an internal comparison between two isoindigos and background with DMF which suggests improved with the Ser modification.

It is suggested that amines have environmental biotoxicity concerns, but the discussion does not directly compare to other amines. In the added reference from Rohr 2013 (Reference 32) a suggestion that MEA, used in carbon capture, may cause breathing irritation. In the added reference from del Rio 2019 (Reference 31) IC50 levels are given for putrescine at 40 mM, where as mM 6AIIIdSer LC50 is 0.2 mM with appropriate conversion. The IC50 and LC50 are not directly comparable making this a tenuous discussion. It is not even clear if putrescine is a relevant compound for carbon capture as the IC50 is from a study on toxicity in food packaging. The isoindigo biocompatibility is not compared to other EMCC sorbents, yet this is listed as one of the 4 main benefit in the conclusion.

In an SDS for MEA from Sigma Aldrich Section 12.1 LC50 for *Cyprinus caprio* (Carp)-349ug/ml

In an SDS for putrescine from Sigma Aldrich Section 11.1 LC50 inhalation rat-1.131ug/ml;
Section 12.1 LC50 *Poecilia reticulata* (guppy)-730ug/ml

In an SDS on isoindigo from CaymanChem there are no currently known LC50s

Given that no isoindigo biotoxicity tests have been given as reference, perhaps this data is still valuable to publish. It appears as though the data is an interesting cytotoxicity test. It does not appear as though these compounds have definitively been shown to be biocompatible. It is encouraged to not list this as a major advantage in the conclusion. Instead, biotoxicity has been explored with a cytotoxicity test. It may be promising for future work, but it is hard to argue “good biocompatibility” has been concluded with the present data.

Response: We are grateful for the suggestions. We have changed the claim of “good biocompatibility” to “improvable biocompatibility” accordingly.

(7) Very minor comment. Fig. 3 R1 and R2 are listed. R1 is the N-substitution modification but it is labeled as simply R and not R1.

Response: Fig. 3 has been revised.

Fig. 3 | Modular synthesis of redox-tunable isoidindigo sorbents for EMCC. Functional groups can be pre-installed onto the precursors of isoidindigos and various isoidindigos can be obtained through Knoevenagel condensation of the precursors. Amino acid ester can be further installed through amidation reaction with carboxylic acid modified isoidindigos.

REVIEWERS' COMMENTS

Reviewer #1 (Remarks to the Author):

The additional comments have been addressed well, and the manuscript has been further improved. I am happy to recommend publication.

Reviewer #3 (Remarks to the Author):

[Note from editor: co-review with reviewer 2]

The changes to the paper in response to all reviewers have been appreciated. It helps the clarity and openness of the research.

While the discussion on biocompatibility is still odd and distracting in the context of the main goals of the paper, the change in wording does make it accurate. No further changes are requested.

This work is well done. Future work on isoindigos will be good to follow.